# Oncogenic *PIK3CA* corrupts growth factor signaling specificity

Ralitsa R Madsen [ID] [1,2 ✉], Alix Le Marois[3,9], Oliwia N Mruk [ID] [2,9], Margaritis Voliotis [ID] [4,9], Shaozhen Yin[1,9], Jahangir Sufi [ID] [5], Xiao Qin [ID] [5,8], Salome J Zhao [ID] [1], Julia Gorczynska [ID] [1], Daniele Morelli [ID] [1], Lindsay Davidson[6], Erik Sahai[3], Viktor I Korolchuk[7], Christopher J Tape[5] & Bart Vanhaesebroeck [ID] [1]

## Abstract

**Technical limitations have prevented understanding of how growth factor signals are encoded in distinct activity patterns of the phosphoinositide 3-kinase (PI3K)/AKT pathway, and how this is altered by oncogenic pathway mutations. We introduce a kinetic, single-cell framework for precise calculations of PI3K-specific information transfer for different growth factors. This features live-cell imaging of PI3K/AKT activity reporters and multiplexed CyTOF measurements of PI3K/AKT and RAS/ERK signaling markers over time. Using this framework, we found that the *PIK3CA^H1047R* oncogene was not a simple, constitutive activator of the pathway as often presented. Dose-dependent expression of *PIK3CA^H1047R* in human cervical cancer and induced pluripotent stem cells corrupted the fidelity of growth factor-induced information transfer, with preferential amplification of epidermal growth factor receptor (EGFR) signaling responses compared to insulin-like growth factor 1 (IGF1) and insulin receptor signaling. *PIK3CA^H1047R* did not only shift these responses to a higher mean but also enhanced signaling heterogeneity. We conclude that oncogenic *PIK3CA^H1047R* corrupts information transfer in a growth factor-dependent manner and suggest new opportunities for tuning of receptor-specific PI3K pathway outputs for therapeutic benefit.**

**Keywords** PI3K Signaling Dynamics; Information Transfer; Growth Factor Specificity; Single-cell Biology
**Subject Categories** Cancer; Signal Transduction

## Introduction

The class IA phosphoinositide 3-kinase (PI3K)/AKT pathway is essential for cellular and organismal homeostasis. It is used for signal transduction downstream of most if not all growth factors (GFs) as well as many hormones and cytokines. The pathway also represents a key therapeutic target due to its frequent hyperactivation across human cancers. This is often due to mutations in *PIK3CA*, the gene encoding the p110α catalytic subunit of the PI3Kα isoform. Based on early cellular studies (Isakoff et al, 2005; Kang et al, 2005; Samuels et al, 2005), common cancer-associated *PIK3CA* mutations such as *PIK3CA^H1047R* are often regarded simply as "on" switches, or activators, of the pathway. Consequently, therapeutic targeting of aberrant PI3K/AKT activation in this context has focused on pathway switch-off (Vanhaesebroeck et al, 2021). However, the efficacy of this approach is often limited by the toxicity of PI3K/AKT inhibition in healthy cells and tissues treated with high doses of PI3K/AKT inhibitors. This is true not only in cancer but also in the non-cancerous *PIK3CA*-related overgrowth spectrum (PROS) of congenital disorders caused by an identical spectrum of activating *PIK3CA* mutations as in cancer (Canaud et al, 2023; Madsen et al, 2018).

The "switch" view of the impact of activating *PIK3CA* mutations and the resulting therapeutic limitations reflect a more general, critical gap in the current knowledge of PI3K/AKT signaling. Specifically, there is limited understanding of how quantitative, dynamic patterns of PI3K/AKT activation are used by cells to specify (or encode) the identity of the myriad environmental signals sensed by this pathway (Madsen and Toker, 2023; Madsen and Vanhaesebroeck, 2020). It therefore also remains unknown if and how disease-causing mutations in PI3K/AKT pathway components may perturb this temporal code. For example, corruption of dynamic signal encoding has been documented in the related RAS/MAPK signaling cascade in response to certain oncogenic BRAF mutations and targeted inhibitors (Bugaj et al, 2018). This type of quantitative mapping of the input-output relationships in the PI3K/AKT pathway is technically challenging. It requires the capture of multimodal biochemical responses with high temporal resolution and quantitative precision (Madsen and Toker, 2023; Madsen and Vanhaesebroeck, 2020). Moreover, unlike conventional protein phosphorylation cascades, the key first step in PI3K pathway

[1]Cell Signaling Laboratory, Department of Oncology, University College London Cancer Institute Paul O'Gorman Building, University College London, London WC1E 6BT, UK. [2]MRC Protein Phosphorylation and Ubiquitylation Unit, School of Life Sciences, University of Dundee, Dundee DD1 5EH, UK. [3]Tumour Cell Biology Laboratory, The Francis Crick Institute, London NW1 1AT, UK. [4]Department of Mathematics and Statistics and Living Systems Institute; University of Exeter, Exeter EX4 4QD, UK. [5]Cell Communication Lab, Department of Oncology, University College London Cancer Institute, London WC1E 6BT, UK. [6]Human Pluripotent Stem Cell Facility, School of Life Sciences, University of Dundee, Dundee DD1 5EH, UK. [7]Biosciences Institute, Faculty of Medical Sciences, Newcastle University, Newcastle upon Tyne NE4 5PL, UK. [8]Present address: MRC Translational Immune Discovery Unit, MRC Weatherall Institute of Molecular Medicine, Oxford OX3 9DS, UK. [9]These authors contributed equally: Alix Le Marois, Oliwia N Mruk, Margaritis Voliotis, Shaozhen Yin. ✉E-mail: rmadsen001@dundee.ac.uk

activation is the generation of the plasma membrane-localized lipid second messenger phosphatidylinositol-3,4,5-trisphosphate (PIP$_3$), and its derivative PI(3,4)P$_2$. The detection of these low-abundance lipids presents a technical challenge, and thus the vast majority of studies of oncogenic PI3K signaling do not feature direct evaluation of this critical first step in PI3K pathway activation, focusing instead on bulk measurements of downstream effector responses (Madsen and Toker, 2023).

Advances in technologies to interrogate single cells have established that cellular heterogeneity in biological responses is the rule rather than the exception (Kramer et al, 2022; Levchenko, 2023; Symmons and Raj, 2016). Thus, quantitative understanding of PI3K signaling in health and disease will also require a shift from bulk to single-cell experimental approaches. This offers the added benefit of enabling the application of information theory to the analysis of biochemical signaling flow, independent of network complexity (Rhee et al, 2012). Information theory is a mathematical framework originally developed by Claude Shannon for the analysis of man-made communication systems. It takes into account not only the magnitude of a response relative to an input, but also the underlying "noise" or uncertainty (Rhee et al, 2012). Therefore, single-cell measurements are key for the application of this framework to cellular signaling. New algorithm developments enable efficient application of information theory not just to individual snapshot responses but also multidimensional signaling trajectories (Jetka et al, 2019). This is crucial given the encoding and decoding of cellular information through signaling dynamics (Madsen and Vanhaesebroeck, 2020; Purvis and Lahav, 2013; Selimkhanov et al, 2014; Sampattavanich et al, 2018; Gross et al, 2019).

Here, we first set out to address the technical limitations that preclude systematic studies of single-cell PI3K/AKT signaling and the associated information transfer at scale. We then used these approaches to study quantitative signal transfer in cell models with allele dose-dependent expression of *PIK3CA*$^{H1047R}$. We discovered that this oncogene corrupted the fidelity of GF-specific information transmission in human cervical cancer (HeLa) cells and in human induced pluripotent stem cells (hiPSCs). This contrasts with the conventional view of *PIK3CA*$^{H1047R}$ as a simple ON switch of the PI3K signaling pathway. Our work now opens for the possibility of using PI3K signaling dynamics as a pharmacological target, a concept first proposed by Behar et al (Behar et al, 2013).

# Results

## Optimized workflow for PI3K activity measurements at the plasma membrane

For quantitative studies of the immediate phosphoinositide lipid outputs of PI3K activation in individual cells and at high temporal resolution, we established a robust, semi-automated live-cell imaging pipeline (Fig. EV1A). Using total internal reflection fluorescence (TIRF) microscopy and the small-molecule PI3Kα activator UCL-TRO-1938 (further referred to as 1938; (Gong et al, 2023)) to monitor PI3Kα outputs specifically at the plasma membrane, we first benchmarked the quantitative fidelity (dynamic range, technical variability) of several pleckstrin homology (PH) domain-based phosphoinositide biosensors (Fig. EV1B). These included the PH domain of BTK (with or without the adjacent TH

domain (Chung et al, 2019; Vihinen et al, 1994)), a tandem-dimer version of the PH domain of ARNO (with modifications to minimize interactions with other proteins (Goulden et al, 2019)), and the PH domain of AKT2 (which is not the full-length protein to avoid internalization independent of PIP$_3$ binding (Ebner et al, 2017)). For consistent comparisons, all biosensors were expressed from the same plasmid backbone that featured a GFP tag at the C-terminus and a nuclear export sequence at the N-terminus (Fig. EV1B). As a control for specificity, all experiments with wild-type (WT) biosensors also featured co-expression of an mCherry-tagged version of each construct, with an arginine-to-alanine mutation that ablates phosphoinositide binding (Fig. EV1B) (Cronin et al, 2004).

The PH domain of AKT2 consistently performed as an optimal biosensor, based on the dynamic range, reproducibility across experiments, low sensitivity to technical noise, and rapid response upon activation as well as inhibition of PI3Kα (Fig. EV1C,D). However, we note that although this sensor measures the total PIP$_3$/PI(3,4)P$_2$ output of PI3K activation at the plasma membrane, the original C-terminal-tagged version of the PH-TH of BTK would be a better option for studies that seek to selectively study PIP$_3$ independently of PI(3,4)P$_2$ (Fig. EV1E). Our final optimized TIRF-based workflow for PI3K activity measurements allowed profiling of up to 4 different cellular conditions (including genotypes) and 60 single cells per experiment, with live-cell measurements taken every 70 s over 60 min while exposing the cells to controlled perturbations, giving rise to more than 3000 individual data points.

## Temporal measurements of class IA PI3K activation identify conserved, dynamic encoding of growth factor signals

Using different cellular model systems (human HeLa cervical cancer cells, human A549 lung adenocarcinoma cells, immortalized mouse embryonic fibroblasts (MEFs)) with or without endogenous functional PI3Kα, we next tested the hypothesis that the identity of different growth factors was captured in the cellular dynamics of PIP$_3$/PI(3,4)P$_2$. Using saturating doses of IGF1 and EGF in serum-free medium, we observed consistent responses that suggested conservation of the dynamic signal encoding of these growth factors across different cell models and species (Fig. 1). At the population level, PIP$_3$/PI(3,4)P$_2$ reporter responses induced by either IGF1 or EGF exhibited a characteristic overshoot, with a peak within the first 10 min of stimulation, followed by a sustained quasi-steady-state above baseline. The key difference between the two growth factors was in the response amplitude, with PI3Kα WT cell lines reaching a peak PIP$_3$/PI(3,4)P$_2$ fold change of ~1.55 for IGF1 and ~1.25 for EGF (Fig. 1).

The combined genetic and pharmacological (BYL719) inactivation of PI3Kα in these experiments also revealed cell type- and growth factor-specific quantitative differences in the contribution of the PI3Kα isoform to each growth factor response (Fig. 1). For example, in HeLa cells, approximately 60% and 50% of the IGF1 and EGF response, respectively, was mediated by PI3Kα. In MEFs, PI3Kα contributed 40% of the IGF1-dependent PIP$_3$/PI(3,4)P2 response, compared to up to 60% of the peak and 50% of the sustained EGF-induced response, respectively. These results therefore suggest that the observed stereotypical IGF1- and EGF-dependent PIP$_3$/PI(3,4)P$_2$ response patterns are robust to the

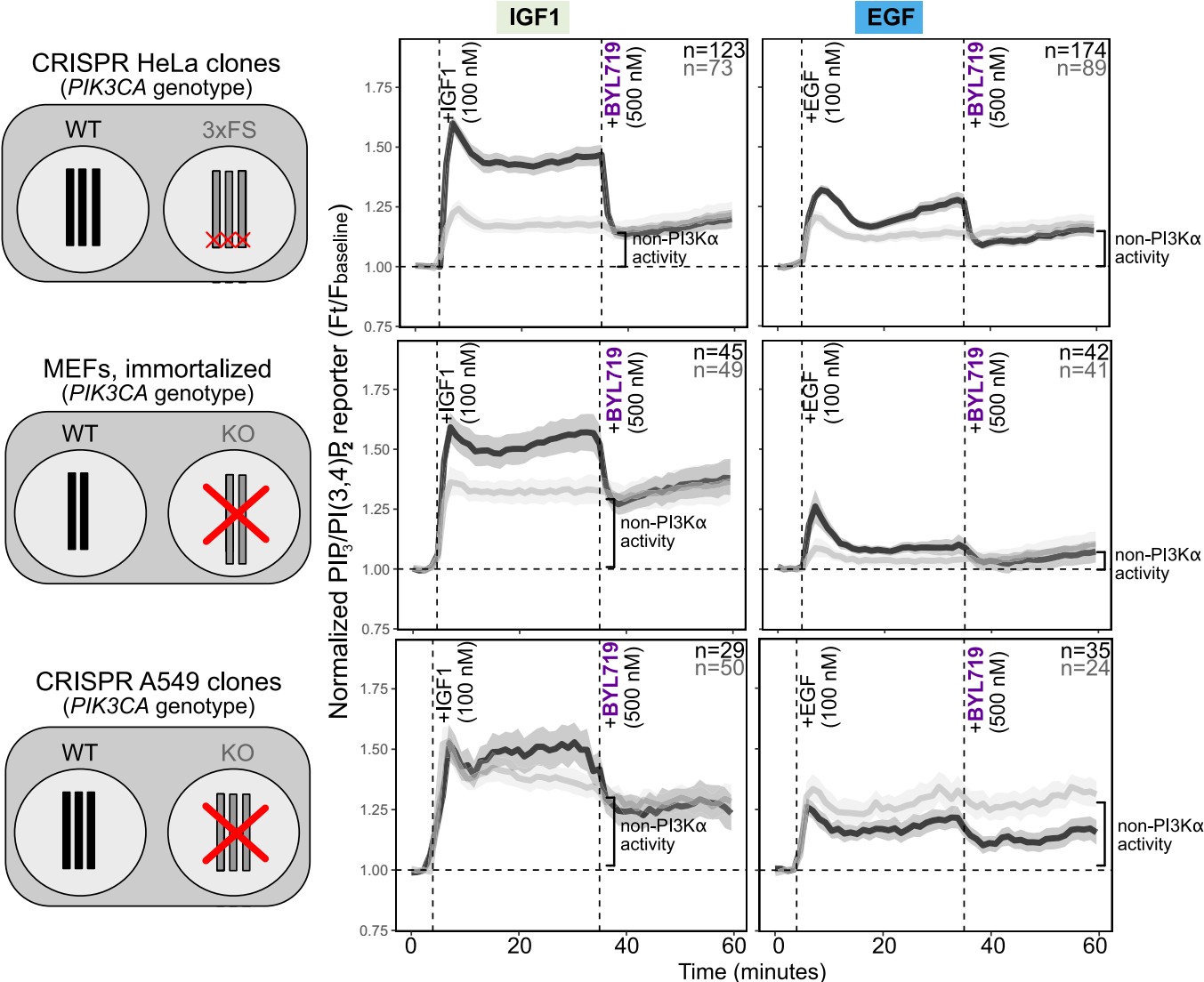

**Figure 1. IGF1 and EGF induce stereotypical PIP₃/PI(3,4)P₂ signaling dynamics.**

Dynamic TIRF microscopy measurements of IGF1- and EGF-induced PIP₃/PI(3,4)P₂ levels in live HeLa, MEF, or A549 cells expressing WT or loss-of-function *PIK3CA* as indicated. The cells were serum-starved for 3 h prior to stimulation with either saturating doses (100 nM) of IGF1 or EGF and treatment with the PI3Kα inhibitor BYL719 (500 nM). The traces represent the mean PH_AKT2 reporter fold change relative to baseline (the median signal of the first four time points). The shading signifies bootstrapped 95% confidence intervals of the mean. The number (*n*) of single cells for each genotype is indicated on the plots. For wild-type (WT) *PIK3CA* HeLa cells, two independent CRISPR/Cas9 clones were used, with and without silent mutations. The 3xFS HeLa cells originate from a single CRISPR/Cas9 clone, engineered with a frameshift (FS) mutation in all three *PIK3CA* alleles (see also Appendix Fig. S1). The MEFs were from polyclonal cultures established from mice with WT or CRE-deleted (KO) *PIK3CA*, followed by immortalization in vitro (Foukas et al, 2010). The A549 cells were from a single CRISPR/Cas9 clone per genotype, with knock-out (KO) of *PIK3CA* caused by a frameshift mutation in exon 3 (Gong et al, 2023). HeLa datasets for IGF1 and EGF are from six and seven independent experiments, respectively. MEF and A549 IGF1 and EGF data are from 3 independent experiments each. Non-PI3Kα activity refers to the PI3K activity that remains following pharmacological inhibition of PI3Kα.

relative contribution of individual class IA PI3K isoforms (Fig. 1). Collectively, these data identify conserved dynamic PI3K-dependent encoding of IGF1 and EGF, with high temporal and isoform-specific resolution.

## Oncogenic *PIK3CA^H1047R^* reduces the PIP₃/PI(3,4)P₂ information capacity of IGF1

We next tested whether the dynamic signal encoding of growth factor identity was equally robust to the expression of *PIK3CA^H1047R^*, one of

the most commonly observed oncogenic PI3Kα mutations in cancer and PROS. Allele dose-dependent, endogenous expression of this variant was engineered in HeLa cells, using CRISPR/Cas9 (Appendix Fig. S1A,B). We chose HeLa cells due to their low baseline PI3K/AKT signaling (Appendix Fig. S1F), absence of pathway-specific mutations, in-depth characterization at multiple biological levels (transcriptomics, proteomics (Bekker-Jensen et al, 2017)), experimental tractability and, in particular, their cervical cancer origin. Genomic profiling of human cervical tumors has revealed these to be among the most enriched for multiple *PIK3CA* mutations in cis or trans (Saito et al, 2020; Sivakumar

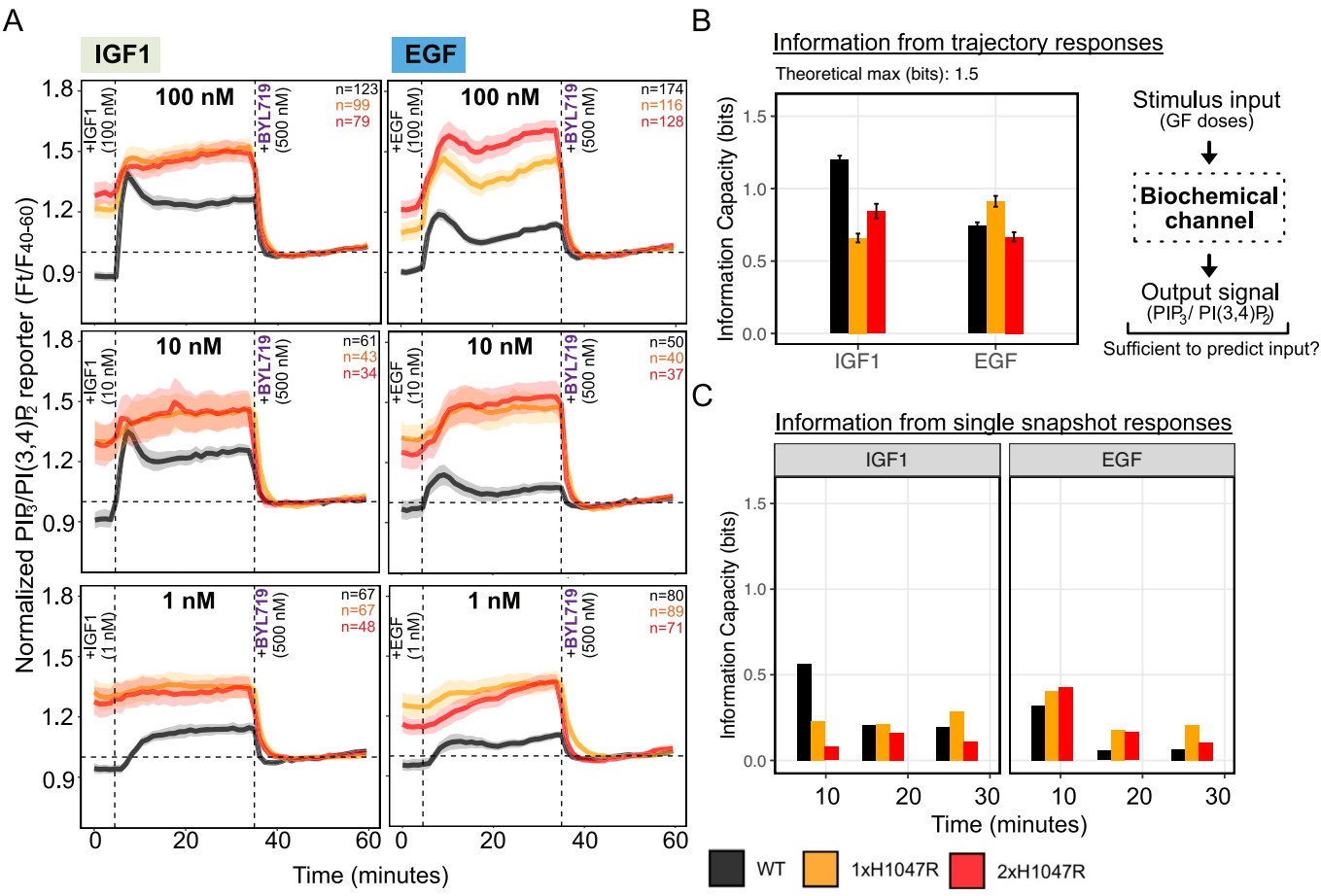

**Figure 2. Oncogenic *PIK3CA^H1047R* in HeLa cells reduces the information capacity in the PIP3/PI(3,4)P2 dynamics for IGF1 but not EGF.**

(A) TIRF microscopy measurements of IGF1- and EGF-induced PIP3/PI(3,4)P2 kinetics in live HeLa cells with endogenous, dose-controlled expression of *PIK3CA^H1047R* (see also Appendix Fig. S1). The cells were serum-starved for 3 h prior to stimulation with the indicated growth factors and treatment with the PI3Kα inhibitor BYL719 (500 nM). Measurements were obtained every 70 s for a total of 60 min. The traces represent the mean PH_AKT2 reporter fold change relative to the median signal for the 40–60 min time window, used here to capture the baseline signaling elevation in *PIK3CA^H1047R* mutant cells. The shaded areas represent bootstrapped 95% confidence intervals of the mean. The shading signifies the 95% confidence intervals of the mean. The number (n) of single cells for each genotype is indicated on the plots. For WT HeLa cells, two independent CRISPR/Cas9 clones were used, with and without silent mutations. The data are from two independent WT, two independent 1xH1047R, and three independent 2xH1047R CRISPR/Cas9 clones. The data are from the following number (n) of independent experiments: n = 6 for 100 nM IGF1; n = 7 for 100 nM EGF; n = 2 for 10 nM IGF1 and 10 nM EGF; n = 3 for 1 nM IGF1; n = 4 for 1 nM EGF. (B) Median information capacity in bits (log2) for IGF1 and EGF calculated from the trajectory responses from all independent experiments and individual cells specified in (A). Capacity is a measure of the maximum amount of information that flows from the pathway input to its output. The theoretical maximum for three inputs (doses) is 1.5 bits if all the information is captured by the PIP3/PI(3,4)P2 dynamics. The height of each bar specifies the median capacity value, with error bars corresponding to the interquartile range (the distance between the first and the third quartiles). These were estimated following 40 bootstrap repetitions using 80% of the initial observations (default diagnostic settings in the SLEMI package (Jetka et al, 2019)). (C) Median information capacity in bits (log2) calculated from snapshot measurements at the indicated time points from the datasets in (A).

et al, 2023). We therefore reasoned that HeLa cells may enable capture of allele dose-dependent effects of *PIK3CA^H1047R* on quantitative signal transfer. So far, such allele dose-dependent effects have only been studied mechanistically in a developmental model system (Madsen et al, 2019, 2021).

Several quality control screens were applied to all final CRISPR/Cas9 clones to identify possible confounders. Assays included whole-exome sequencing (Appendix Fig. S1C), transcriptomics analysis (Appendix Fig. S1D), and candidate-based mRNA and protein expression evaluations (Appendix Fig. S1E,F,G). These assays did not reveal systematic differences across the different clones except for the desired knock-in of *PIK3CA^H1047R* and evidence for an associated yet subtle baseline PI3K/AKT pathway activation by immunoblotting

(Appendix Fig. S1F). This was important because it enabled us to study the consequences of the oncogenic perturbation on signaling response independently of widespread transcriptional changes that could modify the topology of the relevant signaling networks. Finally, we also used a clone with 2xWT and 1xFS *PIK3CA* alleles and confirmed that its PIP3/PI(3,4)P2 reporter response to IGF1 was similar to that in the 3xWT clones (Appendix Fig. S1H). This provides confidence that the CRISPR-generated frameshift *PIK3CA* allele(s) present alongside *PIK3CA^H1047R* do(es) not bias the observed signaling responses.

To assess dynamic signal encoding of IGF1 and EGF as a function of *PIK3CA* genotype, cells were stimulated with one of three different doses (1 nM, 10 nM, 100 nM) of each growth factor, and PIP3/PI(3,4)P2 responses were captured using TIRF

microscopy as described above (Fig. 2A). The resulting temporal measurements of PI3K activity at the single-cell level were processed for mathematical information-theoretic analyses of trajectory responses (Jetka et al, 2019) to quantify the fidelity of dose-dependent signal transfer through PI3K activation. Given three different stimulus doses for each growth factor, the theoretical maximum information capacity captured in the $PIP_3/PI(3,4)P_2$ response would be 1.58 bits ($log2(3)$), corresponding to the case where the $PIP_3/PI(3,4)P_2$ response alone is sufficient to distinguish between the different doses of the growth factor perfectly. Although this is unlikely to be reached given technical noise, values that are substantially lower than 1.58 would imply that the $PIP_3/PI(3,4)P_2$ response alone is not sufficient to distinguish the different doses of each growth factor with high certainty.

We found that HeLa cells expressing WT *PIK3CA* reached a relatively high mean information capacity of 1.2 bits for IGF1, suggesting that the majority of information about the dose of IGF1 was captured in the $PIP_3/PI(3,4)P_2$ response (Fig. 2B). Conversely, there was higher uncertainty about the EGF doses based on the $PIP_3/PI(3,4)P_2$ trajectory alone, with WT *PIK3CA*-expressing HeLa cells reaching a mean information capacity of 0.75 bits. However, expression of the *PIK3CA^H1047R* oncogene resulted in a substantial drop in information transfer downstream of IGF1, particularly in cells expressing a single copy of the mutation. In contrast, single-copy *PIK3CA^H1047R* trended toward increased information capacity downstream of EGF (Fig. 2B).

Three conclusions can be drawn from these data. First, these results demonstrated that oncogenic *PIK3CA^H1047R* could erode signaling fidelity in a growth factor-specific manner. Second, the ability of *PIK3CA^H1047R*-expresing HeLa cells to distinguish between distinct doses of IGF1 on the basis of their $PIP_3/PI(3,4)P_2$ response degraded, reaching similar levels of information capacity as seen for EGF. Third, evaluation of temporal trajectories from the same cells and with high technical precision is key for accurate calculations of information transfer in signaling responses. Information capacity calculations on snapshot measurements from the same data but without the temporal connection revealed erroneously low measures of PI3K signaling fidelity for both growth factors (Fig. 2C).

## *PIK3CA^H1047R* corrupts the specificity of dynamic signal encoding

Further examination of the $PIP_3/PI(3,4)P_2$ trajectories (Fig. 2A) suggested another key difference between HeLa cells expressing WT and *PIK3CA^H1047R*. The EGF-induced $PIP_3/PI(3,4)P_2$ reporter response in mutant cells appeared amplified and largely indistinguishable from that of IGF1 in WT cells. This finding led us to hypothesize that *PIK3CA^H1047R* expression may corrupt the cellular ability to resolve different growth factor inputs from one another. For this to have any effect, however, it would need to be reflected in the activity of key effectors downstream of $PIP_3/PI(3,4)P_2$ generation.

We therefore turned to live-cell imaging of a stably expressed, high-fidelity FOXO-based kinase translocation reporter (KTR) (Gross et al, 2019), whose nucleocytoplasmic distribution provides a proxy measure for AKT activity (Fig. EV2A) and is amenable to high-content-based, quantitative analyses (Fig. EV2B). Compared to TIRF, widefield fluorescence imaging of the FOXO-based KTR response benefits from lower technical noise and allows capture of a

much larger number of individual cells for robust information-theoretic analyses across stimulations with different growth factors. We compared IGF1, insulin, EGF, and epigen due to their paired similarities at the level of activation of distinct receptor tyrosine kinases (RTKs;IGF1R/INSR compared to EGFR). In WT *PIK3CA*-expressing cells, IGF1 and insulin elicited stronger and relatively similar FOXO-based KTR responses compared to EGF and epigen (Figs. 3A and EV2C,D). Moreover, the temporal trajectories of IGF1/insulin KTR responses remained distinct from those of EGF and epigen (Figs. 3A and EV2C,D).

However, simply observing the average trajectories on their own is not enough to determine whether the FOXO-based KTR signaling dynamics are sufficiently distinct to allow individual growth factor inputs to be differentiated from one another. We therefore calculated the mutual information between IGF1 and every other growth factor from the entire set of single-cell trajectories. Mutual information accounts for the probabilistic and thus variable nature of individual growth factor responses. IGF1 was used as the control due to its highly robust single-cell KTR responses (Fig. EV2C,D), both in terms of magnitude and temporal dynamics. *PIK3CA^H1047R*-expressing cells exhibited an allele dose-dependent reduction in mutual information for all growth factors compared to IGF1 (Fig. 3B; mutual information is measured in bits on a log2 scale). This drop was most pronounced for EGF and epigen, in line with the notion of selective amplification of the PI3K/AKT response downstream of EGFR seen in the mutant context. Consequently, the FOXO-based KTR response in mutant cells was no longer sufficiently distinct to resolve different growth factor inputs from one another. We therefore conclude that *PIK3CA^H1047R* corrupts the dynamic encoding of signal identity, giving rise to cells with "blurred biochemical vision" (Fig. 3C).

## *PIK3CA^H1047R* amplifies EGF signaling in cycling cells in 3D culture

A limitation of the approaches presented so far is reliance on exogenous reporters for evaluation of signaling responses. Moreover, TIRF-based measurements of the $PIP_3/PI(3,4)P_2$ reporter response are incompatible with joint tracking of the miniFOXO KTR reporter, limiting analyses to one response at a time. Finally, two-dimensional cell culture models do not capture the biological heterogeneity and additional complexity of three-dimensional (3D) culture systems. We therefore developed an orthogonal approach for single-cell-based signaling measurements in more complex culture settings that retained the ability to perform temporal perturbation experiments at scale.

Specifically, we adapted a highly multiplexed, mass cytometry workflow (Sufi et al, 2021) for use with a new method that we developed for scalable generation of scaffold-free spheroids, including fixation for the preservation of signaling responses and subsequent non-enzymatic single-cell dissociation (Fig. 4A). We used mass cytometry given its versatility, compatibility with cell state-dependent gating and ability to multiplex up to 126 distinct conditions, with gains in sensitivity and technical robustness.

Experiments with saturating doses of IGF1 and EGF revealed that robust growth factor signaling responses in HeLa spheroid cells were restricted to cells that were cycling (pRB Ser^807/811-positive) and non-apoptotic (negative for Caspase 3 cleaved at Asp^175) (Fig. EV3). This

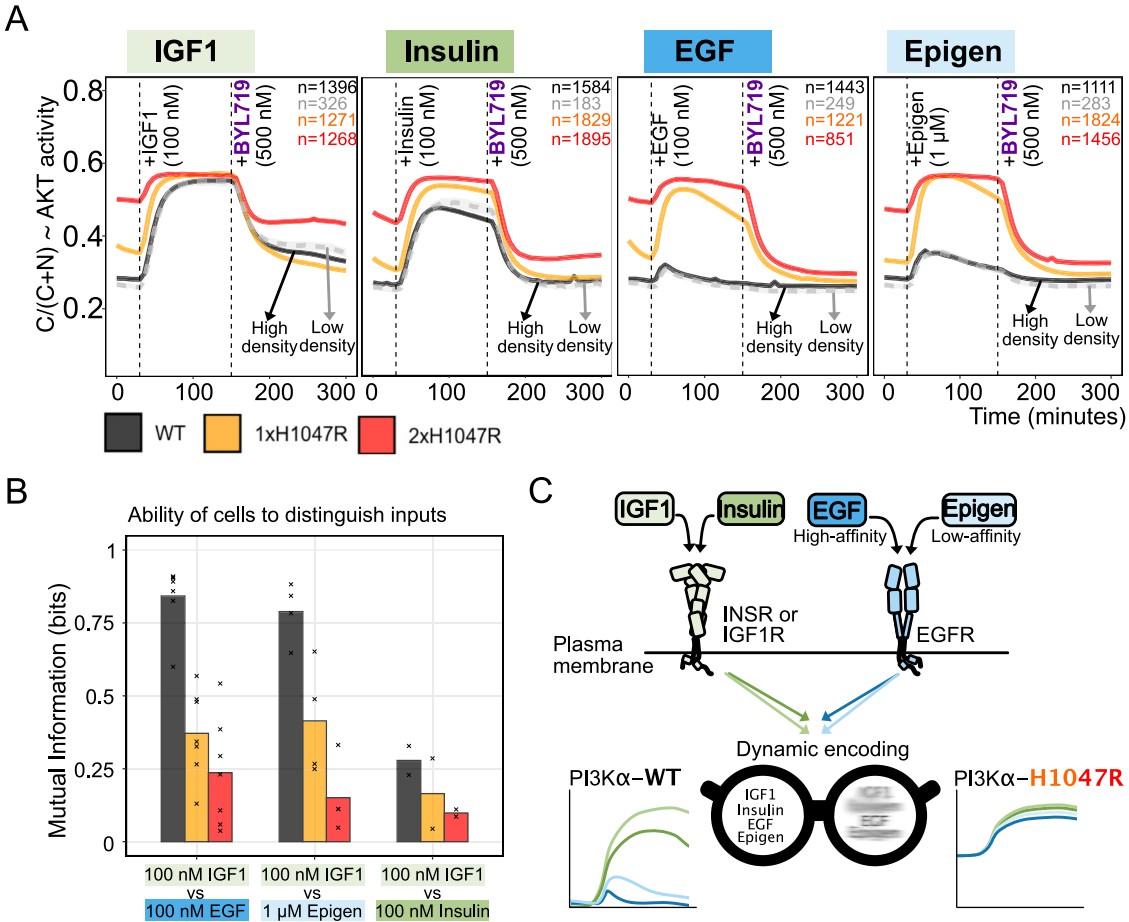

**Figure 3. Oncogenic *PIK3CA*[H1047R] blurs the dynamic encoding of ligand identity in HeLa cells.**

(A) Live-cell fluorescence-based measurements of a miniFOXO-based AKT kinase translocation reporter (KTR) (Gross et al, 2019), stably expressed in HeLa clones with the indicated *PIK3CA* genotypes. The total duration of the time course was 300 min, with measurements obtained every 6 min. For each time point, the traces correspond to the mean proportion of cytoplasmic KTR signal, with shaded areas representing bootstrapped 95% confidence intervals of the mean (note that these may be too small to be seen on the figure). *Y* axis represents an approximation of AKT activity: C, cytoplasmic; N, nuclear. Single-cell numbers (*n*) are shown in the plots. For each condition, WT *PIK3CA*-expressing cells were also seeded at low density to confirm intra-experimental consistency irrespective of cell crowding. The data are representative of a minimum of two independent experiments per condition, performed in two independent CRISPR/Cas9 clones per genotype. Plots from all independent experiments are shown in Fig. EV2 and include control experiments with the 3xFS *PIK3CA* LOF mutant line. (B) Mutual information (MI) in bits (log2) for IGF1 versus each one of the indicated growth factors (EGF, epigen, insulin), calculated using the corresponding KTR trajectory responses (A) prior to inhibitor addition. MI values from individual experimental replicates are indicated as dots overlaid on barplots which correspond to the respective mean of each set of measurements. Because IGF1 gave highly robust KTR dynamics, associated with relatively low single-cell noise as reflected in consistently high MI values, it was chosen as the control stimulus in all experimental replicates. (C) A graphic summarizing the biochemical signal blurring caused by oncogenic *PIK3CA*[H1047R].

finding aligns with human 184A1 breast epithelial cells showing a multimodal, cellular state-conditioned sensitivity to growth factor stimulation (Kramer et al, 2022). However, even when only pRB-positive[+]/cleaved CASP3-negative[-] spheroid cells were examined, the single-cell response distribution shifts relative to control treatments showed discernible, growth factor-specific temporal responses for AKT phosphorylated at Ser[473], ERK1/2 phosphorylated at Thr[202]/Tyr[204] and Thr[185]/Tyr[187] and S6 ribosomal protein phosphorylated at Ser[240/244] (Fig. 4B). For example, in WT *PIK3CA* cells, the single-cell distribution shift for AKT phosphorylation was strongest upon stimulation with 100 nM IGF1 stimulation and peaked after 5–10 min. ERK1/2 phosphorylation in response to EGF showed similar dynamics. Further downstream, a positive distribution shift for S6 phosphorylation followed with a delay relative to phosphorylation of AKT at Ser[473] (Fig. 4B), consistent with prior studies of bulk responses.

Next, to capture the temporal, signaling transitions present in this multidimensional dataset, we used PHATE (potential of heat diffusion for affinity-based transition embedding), which produces a non-linear, low-dimensional embedding that preserves both local and global structure in the data (Moon et al, 2019). We first calculated an earth mover's distance (EMD) score for each response distribution relative to untreated WT control cells; this score measures how different a single-cell distribution for a given signaling marker is relative to another (the corresponding WT control distribution in this case). PHATE was then applied to the EMD scores of all measured signaling markers, which revealed a time-dependent convergence in signaling space of IGF1 and EGF responses for *PIK3CA*[H1047R] mutant cells (Fig. 4C), consistent with the "biochemical blurring" concept (Fig. 3C). This analysis also confirmed a clear distinction between HeLa cells with one versus

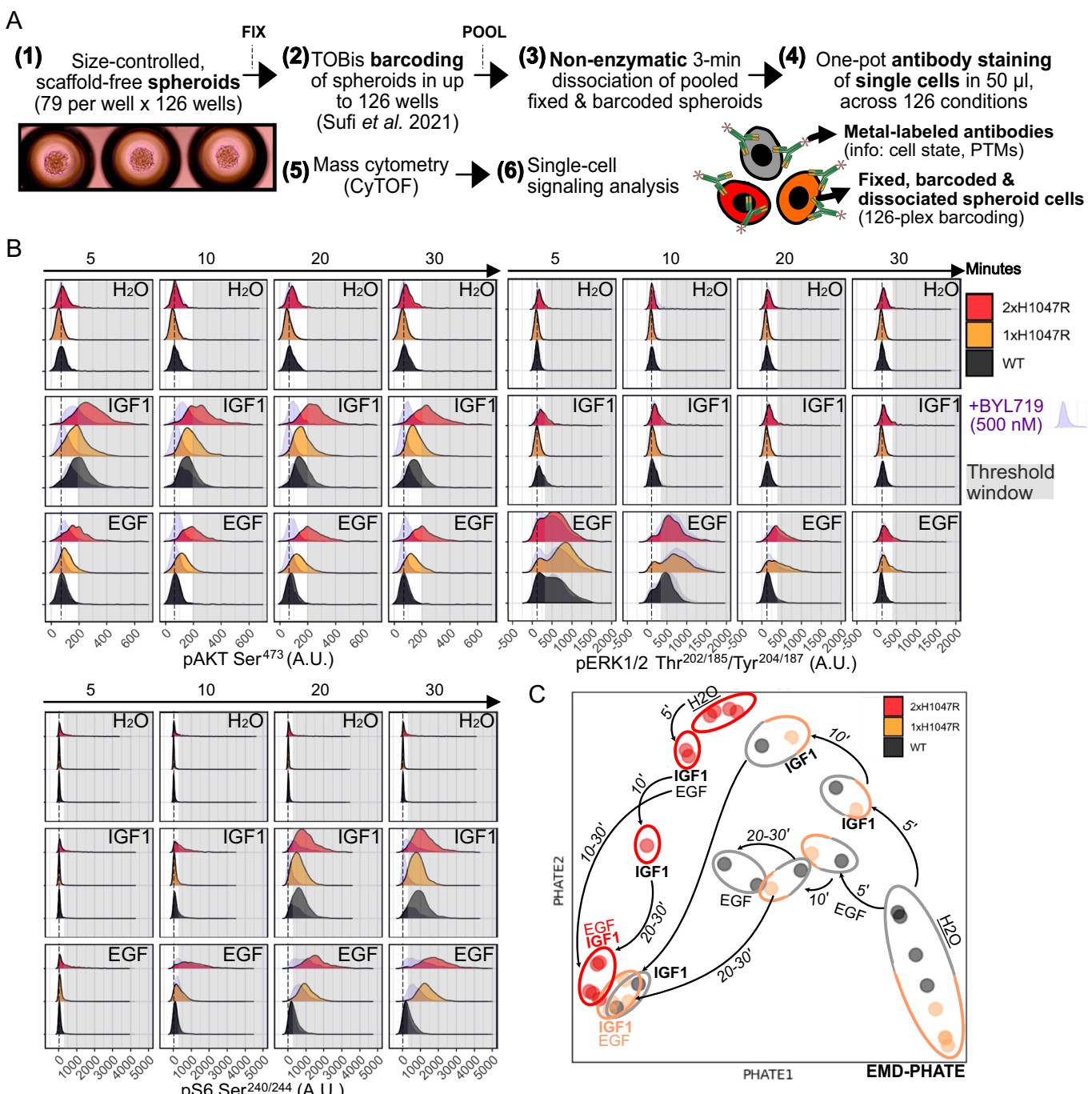

**Figure 4. *PIK3CA^H1047R* amplifies EGF-dependent signaling in a time- and allele dose-dependent manner.**

(A) Overview of the multiplexed mass cytometry (CyTOF) workflow for profiling of single-cell signaling markers in scaffold-free spheroid models. Following fixation and thiol-reactive organoid barcoding in situ (TOBis) (Qin et al, 2020; Sufi et al, 2021), up to 126 conditions are combined into a single sample for non-enzymatic single dissociation which ensures preservation of antibody epitopes, including post-translational modifications (PTMs). Subsequent staining with experimentally validated, metal-conjugated antibodies captures information about cell cycle state (such as cycling, non-cycling, apoptotic) and signaling state. (B) Mass cytometry (CyTOF) data from cycling, non-apoptotic HeLa spheroid cells with endogenous expression of WT *PIK3CA* or one (1xH1047R) or two (2xH1047R) copies of the oncogenic *PIK3CA^H1047R*. The spheroids were serum-starved for 4 h prior to stimulation with 100 nM EGF or IGF1, with and without the PI3Kα inhibitor BYL719 (alpelisib; 500 nM) as a control for signal specificity. Note that BYL719 was added at the same time as the growth factor, not as pre-treatment. The stippled line indicates the position of the peak in WT spheroids treated with vehicle (H₂O). The gray shading highlights the response region not shown by WT *PIK3CA*-expressing cells in the absence of stimulation. (C) Earth mover's distance (EMD)-PHATE embedding of the signaling trajectories observed in the indicated HeLa cell genotypes. Single-cell distributions for the following signaling markers were used for EMD-PHATE processing (see also Fig. EV4): pAKT Ser^473, pERK1/2 Thr^202/Tyr^204; Thr^185/Tyr^187, pNDRG1 Thr^346, pS6 Ser^240/244, pSMAD2/3 Ser^465/467; Ser^423/425.

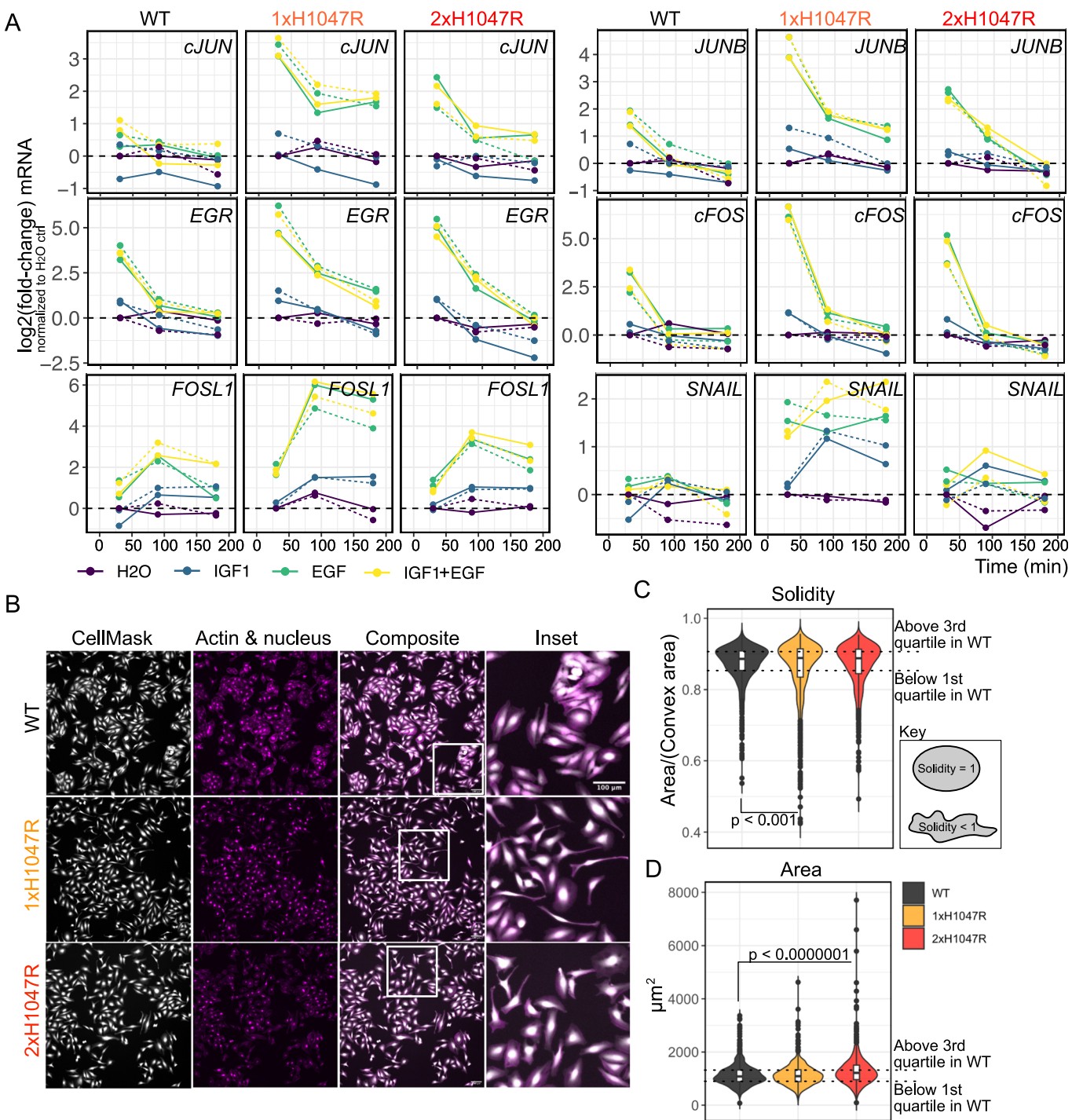

two endogenous copies of *PIK3CA^H1047R*, with the latter also featuring higher baseline levels of the mTORC2 activation marker, NDRG1 phosphorylated at Thr^346 (Fig. EV4A).

In line with our 2D live-cell studies of PI3K/AKT signaling dynamics (Figs. 1 and 2), we also observed that both single and two-copy *PIK3CA^H1047R* mutant cells exhibited an amplified response to EGF stimulation (Figs. 4B,C). The consequence was stronger and more sustained responses in terms of AKT phosphorylated at Ser^473. Similarly, albeit to a lesser degree, the

phosphorylation of ERK1/2 at Thr^202/Tyr^204 and Thr^185/Tyr^187 in *PIK3CA^H1047R*-expressing cells was higher than that in WT controls after 30 min of EGF stimulation, suggestive of a slower switch-off (Figs. 4B,C). Relative to WT *PIK3CA* controls, the PI3K pathway signaling responses in both single- and double-copy *PIK3CA^H1047R* mutant HeLa cells also exhibited increased single-cell variability as a function of time and growth factor stimulation, most notably for the phosphorylation of AKT at Ser^473 (Fig. 4B). This was reproduced with independent clones (Fig. EV4B), and in additional

**Figure 5.** *PIK3CA^{H1047R}* amplifies an EGF-driven transcriptional signature and increases phenotypic diversity in an allele dose-dependent manner.

(A) Bulk transcriptional profiling of EGF-dependent immediate early and delayed early gene expression in HeLa spheroids with endogenous expression of either WT *PIK3CA* or one or two copies of *PIK3CA^{H1047R}* (1-2xH1047R). Expression values are relative and represented as log2 fold-changes, normalized internally to each genotype's control ($H_2O$) response after 30 min of stimulation. All data were further normalized to the expression values of *TBP* (housekeeping gene). The data are representative of two independent experiments (indicated with solid and stippled lines) with one CRISPR-derived clone per genotype. IGF1 and EGF were used at 100 nM, either alone or in combination as indicated. Note the log2 scale of the y axis. (B) Representative fluorescence images of HeLa cells with the indicated genotypes during normal maintenance culture. The cells express a nuclear mCherry marker and were stained with CellMaskBlue and Phalloidin to demarcate their cytoplasm and actin cytoskeleton, respectively. The cells are representative of images from three independent wells per clone and one CRISPR/Cas9 clone per genotype (see also Appendix Fig. S2A for brightfield images of independent HeLa clones for each genotype). The scale bar in (B) corresponds to 100 μm. (C, D) The cytoplasmic images from all replicates were used for deep learning-based segmentation with Cellpose (Stringer et al, 2020). Cell shape solidity (C) and area (D) were quantified for n > 900 single cells per genotype. The P values in (C, D) were calculated according to a one-way ANOVA with Tukey's Honest Significant Difference to correct for multiple comparisons. The exact P value in (C) is 0.00066. The P value in (D) is <0.0000001. The boxplots display five summary statistics. The lower and upper hinges correspond to the first and third quartiles (the 25th and 75th percentiles). The upper whisker extends from the upper hinge to the largest value no further than 1.5 * IQR from the hinge (where IQR is the interquartile range, or distance between the first and third quartiles). The lower whisker extends from the lower hinge to the smallest value at most 1.5 * IQR of the hinge. Data beyond the end of the whiskers are individually plotted outliers.

dose-response, time course experiments using 1 nM, 10 nM, and 100 nM IGF1 or EGF (Fig. EV5A,B). We therefore conclude that oncogenic *PIK3CA^{H1047R}* does not simply shift the PI3K/AKT signaling response to a higher mean but also acts to enhance signaling heterogeneity within the cell population.

## Corrupted signal transfer in *PIK3CA^{H1047R}* mutant cell models translates into increased phenotypic heterogeneity in the context of EGF sensitization

We next set out to test whether the observed signal corruption in *PIK3CA^{H1047R}* mutant cells translates into altered transcriptional and phenotypic responses. First, enhanced EGF signaling through AKT and ERK should lead to an amplification of EGF-specific transcriptional responses, which are sensitive to the relative amplitude and duration of upstream signals such as ERK activation (Avraham and Yarden, 2011; Ram et al, 2023). Consistent with this prediction, we observed increased and more sustained mRNA expression of known EGF-dependent immediately early and delayed early genes in *PIK3CA^{H1047R}* spheroids stimulated with EGF (Fig. 5A). This amplified response was specific to *PIK3CA^{H1047R}* because simply combining saturating concentrations of IGF1 and EGF to elicit strong activation of both AKT and ERK was not sufficient to amplify the transcriptional response in WT *PIK3CA*-expressing cells. Consistent with the role of EGF as an epithelial–mesenchymal transition (EMT)-inducing factor (Cook and Vanderhyden, 2020; Devaraj and Bose, 2019), we also observed increased expression of the EMT-associated transcription factor *SNAIL (SNAI1)* in bulk *PIK3CA^{H1047R}* HeLa spheroids (Fig. 5A). However, the apparent allele dose-dependent pattern of these responses was non-linear, with single-copy *PIK3CA^{H1047R}* mutant cells exhibiting the strongest relative induction of EGF-dependent transcripts known to be associated with EMT downstream of diverse inputs (Cook and Vanderhyden, 2020).

Second, the increased variability in signaling responses in *PIK3CA^{H1047R}* HeLa cells would be expected to cause increased phenotypic heterogeneity. To evaluate this, we visualized cell appearance in standard 2D culture (Fig. 5B). Whereas wild-type HeLa cells grew as epithelial-like cell clusters, single- and double-copy *PIK3CA^{H1047R}* cells were more dispersed and exhibited a higher proportion of cells with irregular, mesenchymal-like morphologies (Fig. 5B–D; Appendix Fig. S2A). The mesenchymal shapes were most pronounced in single-copy *PIK3CA^{H1047R}* mutant cells

(Fig. 5B,C), in line with their higher expression of *SNAIL* upon EGF stimulation (Fig. 5A). Conversely, a higher proportion of the double-copy *PIK3CA^{H1047R}* cells exhibited large, flattened morphologies (Fig. 5B,D).

The phenotypic heterogeneity observed in these HeLa cell models with endogenous, allele dose-dependent *PIK3CA^{H1047R}* expression was similar to that of a previously reported allelic series of non-transformed, human iPSCs with heterozygous and homozygous expression of *PIK3CA^{H1047R}*. Specifically, homozygous *PIK3CA^{H1047R}* iPSC cultures exhibited coexisting epithelial and mesenchymal-like cellular morphologies (Appendix Fig. S2B), consistent with previous findings (Madsen et al, 2019). We hypothesized that this phenotypic heterogeneity reflected corrupted signal transfer in homozygous *PIK3CA^{H1047R}* iPSCs, including amplification of EGF-dependent responses and increased signaling heterogeneity as observed in HeLa cervical cancer cells with 1 or 2 copies of *PIK3CA^{H1047R}*. To test this notion, we screened 3D-cultured, IGF1- or EGF-stimulated iPSCs using our mass cytometry-based, single-cell signaling pipeline (Fig. 4A). As expected, both heterozygous and homozygous *PIK3CA^{H1047R}* iPSCs had higher baseline phosphorylation of AKT at Ser473 relative to WT cells, which increased further upon IGF1 stimulation (Fig. 6A,B). However, only homozygous *PIK3CA^{H1047R}* iPSCs showed an amplified EGF response both at the level of AKT and ERK phosphorylation. This amplified response was accompanied by increased heterogeneity of the underlying *PIK3CA^{H1047R/H1047R}* single-cell responses (Fig. 6A,B), uncovering a conserved signaling phenotype not revealed by conventional workflows based on bulk signaling measurements. In conclusion, compromised—or corrupted—signal transfer downstream of *PIK3CA^{H1047R}* translates into increased signaling and phenotypic heterogeneity in the context of selective amplification of EGF responses (Fig. 6C).

## Discussion

Despite tremendous advances in understanding of the core topology of the PI3K/AKT pathway over the last 30 years, the quantitative mechanisms of signal-specific information transfer in this pathway have remained elusive due to technical and analytical limitations (Madsen and Toker, 2023; Madsen and Vanhaeseb-roeck, 2020). In this work, we present a suite of optimized single-cell-based, kinetic workflows for systematic mapping of

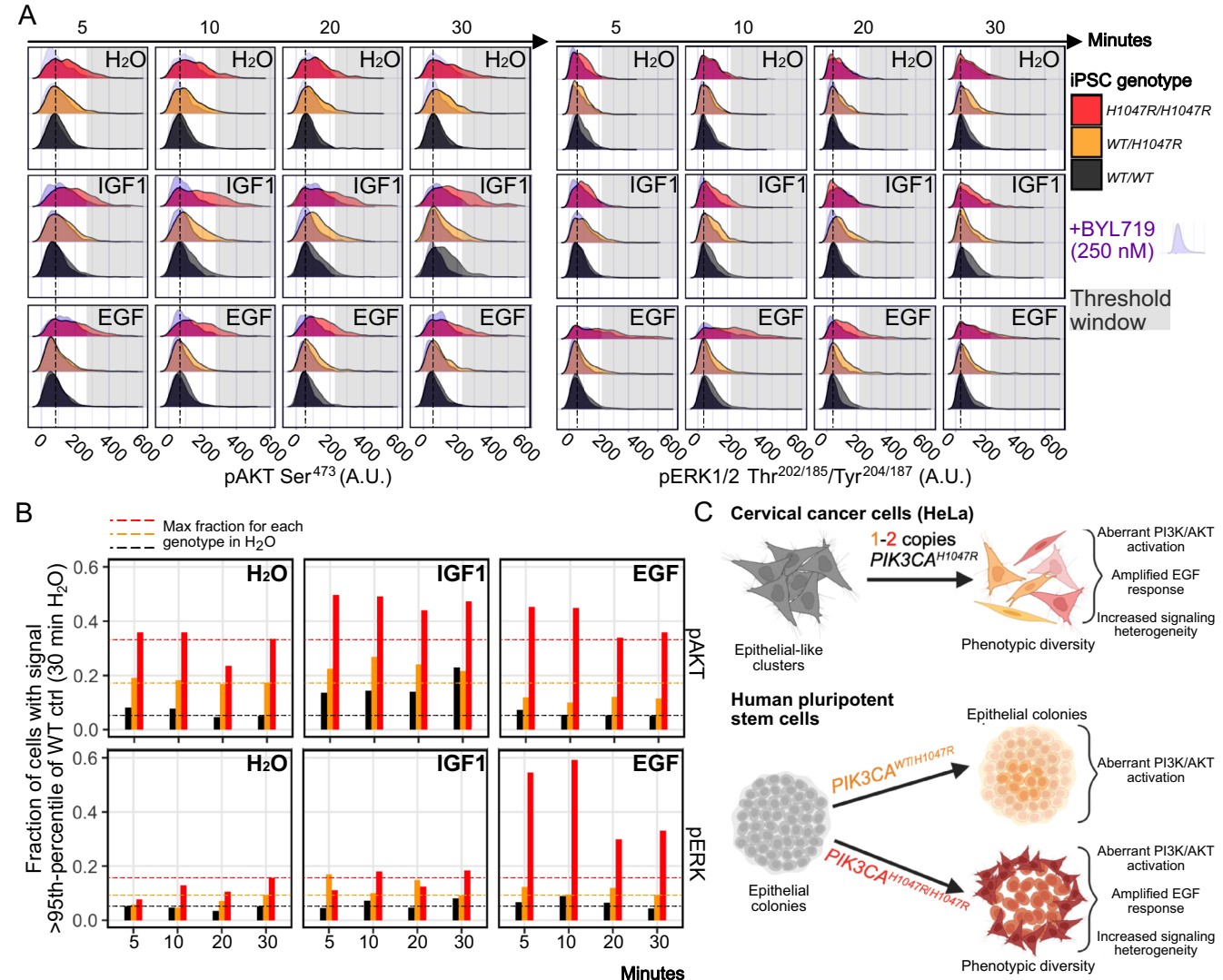

**Figure 6.  Corrupted signal transfer and EGF response amplification is conserved in homozygous *PIK3CA^H1047R* iPSCs.**

(**A**) Mass cytometry data from cycling (pRB+) iPSC spheroid cells with WT *PIK3CA* or heterozygous or homozygous *PIK3CA^H1047R* expression. The spheroids were serum-starved for 2 h prior to stimulation with 100 nM EGF or IGF1, with and without the PI3Kα inhibitor BYL719 (alpelisib; 250 nM) as a control for signal specificity. Note that BYL719 was added at the same time as the growth factor, not as pre-treatment. The stippled line indicates the position of the peak in WT spheroids treated with vehicle (H$_2$O). The gray shading highlights the response region not shown by WT *PIK3CA*-expressing cells in the absence of stimulation. (**B**) Thresholding of the data in (**A**) to quantify the percentage of cells within each condition with a pAKT or pERK signal above the corresponding 95th percentile of vehicle (H$_2$O)-treated WT iPSCs at 30 min. The stippled lines indicate the maximum fraction of cells within this threshold for each genotype prior to growth factor stimulation. The data are from a confirmatory screen with one iPSC clone per genotype and n > 260 single cells per condition. (**C**) Graphical summary of the key observations of the impact of *PIK3CA^H1047R* expression in HeLa and iPSC cells. Created with BioRender (https://BioRender.com/v54q309).

quantitative signaling specificity in PI3K/AKT pathway activation. Supported by information-theoretic analyses, we show that endogenous expression of the *PIK3CA^H1047R* cancer hotspot variant results in quantitative blurring of growth factor-specific information transfer, amplification of EGF-induced responses and increased phenotypic heterogeneity in an allele dose-dependent manner. We note that an early study of bulk PI3K signaling responses in non-transformed breast epithelial cells also observed sensitization to EGF in the presence of either *PIK3CA^H1047R* or the helical domain hotspot variant *PIK3CA^E545K* (Gustin et al, 2009). Our quantitative framework now allows these results to be contextualized into a coherent model of growth factor-specific mechanisms of action of oncogenic *PIK3CA*.

We propose a model in which oncogenic *PIK3CA^H1047R* is not a simple ON switch of the PI3K/AKT pathway but acts as a context-dependent signal modifier, determined by the principles of quantitative biochemistry (Madsen and Toker, 2023; Nussinov et al, 2022). Accordingly, the most parsimonious explanation for the selective amplification of EGF-dependent responses by *PIK3CA^H1047R* is the ability of this mutation to increase p110α residency times at lipid membranes (Burke et al, 2012; Jenkins et al, 2023, p. 202), alongside the lack of high-affinity phospho-tyrosine

(pTyr) binding sites for the regulatory p85 subunit on EGFR and its associated adaptor proteins (Gordus et al, 2009). This would make the interaction between p110α and RAS essential for efficient PI3K-dependent signal transduction downstream of the EGFR (Gupta et al, 2007; Rodriguez-Viciana et al, 1994; Vadas et al, 2011; Wennström and Downward, 1999), which would explain two key observations in our data. The first is the substantial increase in EGF-induced ERK phosphorylation in cell lines expressing only mutant *PIK3CA^{H1047R}*. The second is the complete suppression of AKT KTR responses following PI3Kα-selective inhibition in the context of EGF but not IGF1/insulin stimulation (Figs. 3 and Fig. EV2). The PIP$_3$/PI(3,4)P$_2$ and AKT/FOXO (KTR) trajectories showed that IGF1 (and insulin) signal through both PI3Kα and an additional class IA PI3K isoform. This is unsurprising given the ability of p85 to bind efficiently to pTyr (pYxxM) sites on IGF1R and insulin receptor substrate (IRS) proteins irrespective of the catalytic p110 subunit (Luo et al, 2005; Tsolakos et al, 2018). It would also explain the convergence of IGF1/insulin-induced AKT/FOXO (KTR) trajectories in *PIK3CA* loss-of-function and *PIK3CA^{H1047R}* cells following PI3Kα-selective inhibition (Fig. EV2). The only cell line in this work in which PI3Kα appeared dispensable for EGF-induced PI3K signaling was A549, a lung adenocarcinoma cell line with amplified EGFR (Greshock et al, 2008). This aligns with the above biochemical considerations because a higher concentration of EGFR would allow for more successful engagement of low-affinity interactors such as p85, thus reducing the dependence of PI3K activation on RAS binding through p110α. This follows from the inability of p110β, which is the other main catalytic p110 isoform in the non-hematopoietic cell lineages used here, to directly interact with RAS (Burke and Williams, 2015).

The above model is nevertheless a simplification because it does not account for another salient property of oncogenic *PIK3CA*, revealed by our quantitative single-cell measurements of PI3K/AKT signaling dynamics. Thus, we consistently observed an increase in signaling and phenotypic heterogeneity downstream of allele dose-dependent *PIK3CA^{H1047R}* expression. It is interesting that an increased heterogeneity in PI3K pathway activation was also noted in an early study of *PIK3CA^{H1047R}* overexpression in breast epithelial cells (Yuan et al, 2011). The consequences of this heterogeneity are two-fold. First, it increases the uncertainty in the ability to predict the outputs of oncogenic PI3K/AKT pathway activation, meaning that the outputs are probabilistic rather than deterministic. This calls for increased attention to single-cell PI3K signaling responses in the ongoing evaluation of the many PI3K/AKT pathway inhibitors entering preclinical and clinical use (Castel et al, 2021; Vanhaesebroeck et al, 2021, 2022). Second, such heterogeneity endows cells with the ability to sample multiple phenotypic states or attractors (Feinberg and Levchenko, 2023), as shown by the emergence of coexisting cellular phenotypes in otherwise isogenic cells with *PIK3CA^{H1047R}* expression. This may offer a mechanistic underpinning for the phenotypic heterogeneity found in *PIK3CA*-driven breast cancer models (Hanker et al, 2013; Koren et al, 2015; Van Keymeulen et al, 2015) as well as in benign but highly debilitating human PROS disorders (Madsen et al, 2018). It is likely that the observed increase in single-cell heterogeneity endows the population of mutant *PIK3CA^{H1047R}* cells with a selective advantage in the face of unpredictable and rapidly changing environments, as shown in other systems (Suderman et al, 2017).

Finally, our quantitative PI3K signaling framework can now be used for screening purposes to identify therapeutic modalities that normalize growth factor-specific signal transfer in *PIK3CA^{H1047R}* cells. This would be different from direct PI3Kα inhibition which, as suggested by our genetic and pharmacological data, does not restore normal signaling dynamics. Given our evidence for conserved receptor-specific mechanisms of PI3K-dependent signal encoding, alternative pharmacological approaches may instead feature modulation of receptor-specific adaptor proteins, phosphatases and/or E3 ligases, all of which are critical for dynamic signaling control through regulation of receptor internalization, recycling and degradation (Madsen and Toker, 2023). Similar modulation has also been suggested for RAS/MAPK signaling (Bivona, 2019), in light of the finding that oncogenic mutations in this pathway also remain dependent on upstream growth factor inputs yet fail to transmit these reliably (Bugaj et al, 2018). Given the likely dependence on a direct PI3Kα-RAS interactions for the EGF response amplification in *PIK3CA^{H1047R}* cells, the recently developed PI3Kα-RAS breaker (ClinicalTrials.gov ID NCT06625775) may be an excellent candidate for testing of quantitative, growth factor-specific PI3K signaling dynamics as a pharmacological target.

# Methods

**Reagents and tools table**

| Reagent/resource | Reference or source | Identifier or catalog number |
| --- | --- | --- |
| **Experimental models** | | |
| Human cervical cancer (HeLa) cells | ATCC | #CCL-2 |
| Immortalized mouse embryonic fibroblasts (MEFs) with wild-type PIK3CA and PIK3CA knock-out | Foukas et al, 2010 | |
| Human lung adenocarcinoma A549 cells | ATCC | #CCL-185 |
| WTC11 line (RRID: CVCL_Y803) with wild-type PIK3CA, PIK3CA^{WT/H1047R} or PIK3CA^{H1047R/H1047R} | Madsen et al, 2019 | Parental line: RRID: CVCL_Y803 |
| **Recombinant DNA** | | |
| pSBbi-FoxO1_1R_10A_3D (designated MB40) | Addgene | RRID:Addgene_106278 |
| pCMV(CAT)T7-SB100 (designated MB43) | Addgene | RRID:Addgene_34879 |
| pNES-EGFP-C1-PH-ARNO(I303E)x2 | Addgene | RRID:Addgene_116868 |
| pNES-mCherry-C1-TAPP1-cPHx3 | Addgene | RRID:Addgene_116855 |
| PH-BTK-GFP (original) | Addgene | RRID:Addgene_51463 |
| pNES-mCherry-C1-PH-mutARNO-R280A-2G-I303Ex2 | MRC Reagents & Services | #DU78205 |
| pNES-EGFP-C1-PH-mutARNO-R280A-2G-I303Ex2 | MRC Reagents & Services | #DU78206 |

| Reagent/resource | Reference or source | Identifier or catalog number |
|---|---|---|
| pNES-EGFP-C1-PH-BTK-WT | MRC Reagents & Services | #DU78207 |
| pNES-mCherry-C1-PH-BTK-MUT | MRC Reagents & Services | #DU78208 |
| pNES-EGFP-C1-PH-AKT2-WT | MRC Reagents & Services | #DU78209 |
| pNES-mCherry-C1-PH-AKT2-MUT | MRC Reagents & Services | #DU78210 |
| pNES-GFP-PH-TH-BTK | MRC Reagents & Services | #DU78211 |
| pNES-GFP-PHmut-TH-BTK | MRC Reagents & Services | #DU78212 |
| pUC19 | New England Biolabs | #09052008 |
| **Antibodies** | | |
| Primary antibodies for Western blotting | See Appendix Table S1 | |
| Secondary antibodies for Western blotting | See Appendix Table S2 | |
| Primary antibodies for mass cytometry | See Appendix Table S3 | |
| **Oligonucleotides and other sequence-based reagents** | | |
| HDR001 | Madsen et al, 2019 and this study | |
| HDR002 | Madsen et al, 2019 and this study | |
| sgRNA for PIK3CA exon 21 | Madsen et al, 2019 and this study | Synthego EZ sgRNA custom order |
| PIK3CA exon 21 Sanger sequencing primers | Madsen et al, 2019 and this study | Sigma Aldrich oligonucleotide custom order |
| MiSeq PIK3CA exon 21 primers | This study | Sigma Aldrich oligonucleotide custom order |
| BTK PH domain cloning and site-directed mutagenesis primers | This study | Sigma Aldrich oligonucleotide custom order |
| AKT2 PH domain cloning and site-directed mutagenesis primers | This study | Sigma Aldrich oligonucleotide custom order |
| RT-qPCR primers | See Appendix Table S4 | |
| **Chemicals, enzymes, and other reagents** | | |
| Nunc™ Cell-Culture Treated Multidishes | Thermo Fisher Scientific | #140675 |
| Corning 25 cm² Rectangular Canted Neck Cell Culture Flask with Vent Cap | Corning | #430639 |
| DMEM high glucose (with 4 mM L-Glutamine and 1 mM sodium pyruvate) | Thermo Fisher Scientific | #41966-029 |
| L-Glutamine | Sigma Aldrich | #G7513 |
| 10% fetal bovine serum | Pan-Biotech | #P30-8500 |
| RPMI-1640 with GlutaMax and Sodium Bicarbonate | Thermo Fisher Scientific | #61870-036 |

| Reagent/resource | Reference or source | Identifier or catalog number |
|---|---|---|
| Sodium Pyruvate | Thermo Fisher Scientific | #11360-039 |
| DPBS | Sigma Aldrich | #RNBH8966 |
| DPBS | Thermo Fisher Scientific | #14190-094 |
| TrypLE™ Express Enzyme | Thermo Fisher Scientific | #12605028 or #1260421 |
| DMEM/F12 | Thermo Fisher Scientific | #21331-046 |
| Essential 8 Flex Medium | Thermo Fisher Scientific | #A2858501 |
| Cultrex Stem Cell Qualified Reduced Growth Factor Basement Membrane | R&D Systems | #3434-010-02 |
| 10 μM Y-27632 dihydrochloride | Bio-Techne | #1254/10 |
| StemPro Accutase | Thermo Fisher Scientific | #A1110501 |
| Alt-R™ S.p. Cas9 Nuclease V3 | IDT | #1081061 |
| SE Cell Line 4D-Nucleofector™ X Kit S | Lonza | #V4XC-1032 |
| QuickExtract | Cambridge Bioscience | #QE0905T |
| NucleoSpin Micro Kit XS | Takara | #740901.5 |
| Next Ultra DNA Library Prep Kit | New England Biolabs | #E7370L |
| Agilent's SureSelectXT Reagent Kit and Agilent SureSelect Human All ExonV6 | Agilent | #G9611B |
| Fugene HD Transfection Reagent | Promega | #E5912 |
| Opti-MEM I Reduced Serum Medium | Thermo Fisher Scientific | #31985070 |
| Puromycin | Sigma Aldrich | #P9620 or #P4512-1MLX10 |
| RIPA Lysis and Extraction buffer | Thermo Fisher Scientific | #89900 |
| cOmplete™, Mini, EDTA-free Protease Inhibitor Cocktail | Sigma Aldrich | #4693159001 |
| cOmplete ULTRA Tablets, Mini, EasyPack, PhosStop | Sigma Aldrich | #4906845001 |
| 4–12% Bis-Tris Midi NuPage Protein Gels | Thermo Fisher Scientific | #WG1402BOX or #WG1401BOX |
| NuPAGE LDS Sample Buffer (4X) | Thermo Fisher Scientific | #NP0007 |
| NuPAGE Sample Reducing Agent (10X) | Thermo Fisher Scientific | #NP0009 |
| NP MES SDS Running Buffer 20X | Thermo Fisher Scientific | #NP0002 |
| NuPAGE Antioxidant | Thermo Fisher Scientific | #NP0005 |
| Immobilon Forte Western HRP substrate | Sigma Aldrich | #WBLUF0500 |

| Reagent/resource | Reference or source | Identifier or catalog number |
|---|---|---|
| ECL Western Blotting Substrate | Promega | #W1015 |
| Human IGF1 | Peprotech | #100-11 |
| Human EGF | Peprotech | #AF-100-15 |
| Human Epigen | Peprotech | #100-51 |
| Human insulin (10 mg/ml) | Sigma Aldrich | #91077 C |
| BYL719 | SelleckChem | #S2814 |
| TGX221 | MedChemExpress | #HY-10114 |
| 1938 | CancerTools | #161068 |
| Ambion non-DEPC-treated water | Thermo Fisher Scientific | #9937 |
| Sterile DMSO | Cell Signaling Technology | #12611S |
| Sterile DMSO | Sigma Aldrich | #D2650 |
| BamHI-HF | New England Biolabs | #R3136S |
| HindIII-HF | New England Biolabs | #R3104S |
| Quick alkaline phosphatase (calf intestinal) | New England Biolabs | #M0525S |
| rCutSmart buffer | New England Biolabs | #B6004S |
| Monarch DNA gel extraction kit | New England Biolabs | #T020S |
| 2X instant sticky-end ligase master mix | New England Biolabs | #M0370S |
| High-efficiency 5-alpha competent *E. coli* | New England Biolabs | #C2987I |
| Maxi Plus kit | Qiagen | #12964 |
| Platinum SuperFi DNA Polymerase | Thermo Fisher Scientific | #12351-010 |
| Corning® Elplasia® 24-well Black/Clear Round Bottom Ultra-Low Attachment, Microcavity (554) | Corning | #4441 |
| Corning® Elplasia® 96-well Black/Clear Round Bottom Ultra-Low Attachment plates, Microcavity (79) | Corning | #4442 |
| μ-Dish 35 mm, high Glass Bottom; TIRF / superresolution applications | Ibidi | #81158 |
| 25 Culture-Inserts 4 Well for self-insertion | Ibidi | #80469 |
| Luer Lock Connector, Female | Ibidi | #10825 |
| PFTE tubing (Diba Omnifit Tubing, PTFE, 1/16" (1.6 mm) OD x 0.5 mm ID; 20 m/pk) | Cole-Parmer | #WZ-21942-70 |
| Pharmed tubing | Biorad | #7318208 |
| 3 ml syringes | Terumo | #MDSS03SE |
| Growth Factor-reduced Matrigel | Corning | #354230 |
| Penicillin-Streptomycin | Sigma Aldrich | #P4333 |

| Reagent/resource | Reference or source | Identifier or catalog number |
|---|---|---|
| Fluorobrite DMEM | Thermo Fisher Scientific | #A1896701 |
| 16% methanol-free formaldehyde | Polysciences | #18814-20 |
| Perkin Elmer ViewPlate-96 dishes (TC-treated) | Perkin Elmer | #6005182 |
| High-Capacity cDNA Reverse Transcription Kit | Thermo Fisher Scientific | #4368814 |
| PowerUP™ SYBR™ Green Master Mix | Thermo Fisher Scientific | #A25742 |
| 5X fish skin gelatin blocking agent | Biotium | #22010 |
| Tween® 20, Molecular Biology Grade | Promega | #H5152 |
| Triton® X-100, Molecular Biology Grade | Promega | #H5141 |
| Phalloidin iFlouor 555 | abcam | #ab176756 |
| CellMaskBlue | Thermo Fisher Scientific | #H32720 |
| Tissue Grinder Dissociation Tubes with 40 μm strainer | Fast Forward Discoveries | Custom order |
| L-Glutathione (reduced) | Sigma Aldrich | #G6529-5G |
| EDTA 0.5 M stock solution, pH = 8 | Sigma Aldrich | #03690-100 ml |
| Cell-IDTM Intercalator-Ir | Standard BioTools | #201192A |
| Maxpar Cell Staining Buffer | Standard BioTools | #201068 |
| Maxpar Water | Standard BioTools | #201069 |
| Maxpar PBS | Standard BioTools | #201058 |
| Fix & Perm buffer | Standard BioTools | #201067 |
| Maxpar Cell Acquisition Solution Plus for CyTOF XT | Standard BioTools | #201244 |
| EQ Six Element Calibration Beads | Standard Biotools | #201245 |
| TOBis barcodes | Sufi et al, 2021 | |
| **Software** | | |
| R | https://www.r-project.org | |
| Python | https://www.python.org | |
| Synthego's ICE pipeline | Hsiau et al, 2019 | |
| CRISPResso2 pipeline | Clement et al, 2019 | |
| Nextflow (version 20.07.1) | https://www.nextflow.io | |
| nf-core RNAseq pipeline (v1.1) | Ewels et al, 2020 | |
| nf-core/sarek pipeline (v2.6.1) | https://nf-co.re/sarek/2.6.1/ | |
| Strelka2 pipeline | Kim et al, 2018 | |
| snpEff pipeline | Cingolani et al, 2012 | |
| Spliced Transcripts Alignment to a Reference (STAR) pipeline | Dobin et al, 2013 | |

| Reagent/resource | Reference or source | Identifier or catalog number |
|---|---|---|
| featureCounts pipeline | Liao et al, 2014 | |
| limma R package | Ritchie et al, 2015 | |
| ComplexHeatmap R package | Gu et al, 2016 | |
| TCGAbiolinks R package | https://doi.org/10.18129/B9.bioc.TCGAbiolinks | |
| PCAtools package | https://doi.org/10.18129/B9.bioc.PCAtools | |
| Slidebook 6.0 | https://www.intelligent-imaging.com/slidebook | |
| NIS Elements Software V.5.43.03 | Nikon | |
| Fiji open-source image processing package | Schindelin et al, 2012 https://imagej.net/software/fiji/ | |
| LabelsToROIs plugin | https://labelstorois.github.io | |
| Time Course Inspector package | Dobrzynski et al, 2019 | |
| Cellpose (v1) | Stringer et al, 2020 | |
| Stardist | Schmidt et al, 2018 | |
| Trackmate | Tinevez et al, 2017 | |
| CyTOF debarcoding software | Zunder et al, 2015 | |
| CyGNAL package | Sufi et al, 2021 | |
| flowCore | Hahne et al, 2009 | |
| CATALYST | Crowell et al, 2023 | |
| ggridges | https://cran.r-project.org/web/packages/ggridges/index.html | |
| tidyverse | https://www.tidyverse.org | |
| SLEMI (Statistical Learning-based Estimation of Mutual Information) R package was used | Jetka et al, 2019 | |
| jupyter | https://jupyter.org | |
| pandas 1.5.3 | https://pandas.pydata.org | |
| numpy 1.24.4 | https://numpy.org | |
| scprep 1.2.3 | https://github.com/KrishnaswamyLab/scprep | |
| scipy 1.11.1 | https://scipy.org | |
| matplotlib 3.7.2 | https://matplotlib.org | |
| fcsparser 0.2.6 | https://github.com/eyurtsev/fcsparser | |
| seaborn 0.12.2 | https://seaborn.pydata.org | |

| Reagent/resource | Reference or source | Identifier or catalog number |
|---|---|---|
| PHATE | Moon et al, 2019 | |
| BioRENDER | https://app.biorender.com | |
| **Other** | | |
| iBlot2 system | Thermo Fisher Scientific | |
| Amersham ImageQuant 800 system | Cytiva | |
| NovaSeq 6000 | Illumina | |
| MiSeq | Illumina | |
| Helios or XT mass cytometer | Standard BioTools | |
| Quant Studio™ 6 Real-Time PCR System | Thermo Fisher Scientific | |
| Ti2 Eclipse microscope | Nikon | |
| 3i Spinning Disk Confocal and TIRF microscope | 3i | |
| sCMOS Prime95B (Photometric) | Photometric | |
| CMOS Prime BSI | Photometric | |
| 100 × 1.45 NA plan-apochromatic oil-immersion TIRF | Zeiss | |
| 10 × 0.45 NA plan-apochromatic dry objective | Nikon | |
| Tissue Grinder Unit | Fast Forward Discoveries | |

## Immortalized cell culture

Human cervical cancer (HeLa) cells (ATCC #CCL-2 and CRISPR derivatives) and previously established immortalized mouse embryonic fibroblasts (MEFs; generated in Foukas et al, 2010) were cultured in complete medium consisting of DMEM (with 4 mM L-Glutamine and 1 mM sodium pyruvate; Thermo Fisher Scientific #41966-029) supplemented with an additional 2 mM of L-Glutamine (Sigma #G7513) and 10% fetal bovine serum (FBS; Pan-Biotech #P30-8500). Human lung adenocarcinoma A549 cells (ATCC #CCL-185 and CRISPR derivatives generated in ref. (Gong et al, 2023)) were cultured in complete medium consisting of RPMI-1640 with GlutaMax and Sodium Bicarbonate (#61870-036, Thermo Fisher Scientific), supplemented with 1 mM of Sodium Pyruvate (#11360-039, Thermo Fisher Scientific) and 10% FBS. Cells were cultured in T25 flasks (Corning or TPP) and passaged every two-to-three days when 80–90% confluent. Briefly, the spent medium was removed and the cells washed with 5 ml DPBS (Sigma #RNBH8966 or Thermo Fisher Scientific #14190-094). Following removal of the wash, the cells were incubated at 37 °C in 0.75 ml TrypLE™ Express Enzyme (Thermo Fisher Scientific #12605028 or #1260421) for 6–8 min until dissociated. The cells were resuspended in complete medium and distributed to new flasks at appropriate ratios.

## Human-induced pluripotent stem cell culture

The male human iPSCs used in this work were derived from the commercially available WTC11 line (RRID: CVCL_Y803) and were previously engineered by CRISPR/Cas9 to endogenously express *PIK3CA*<sup>H1047R</sup> (reported and characterized in Madsen et al, 2019). The cells were maintained in Essential 8 Flex Medium (Thermo Fisher Scientific #A2858501) on plates coated with 10 µg/cm² Cultrex Stem Cell Qualified Reduced Growth Factor Basement Membrane (R&D Systems #3434-010-02). Cells were cluster-passaged every 3–4 days with 0.5 mM EDTA and seeded into medium supplemented with 10 µM Y-27632 dihydrochloride (Bio-Techne #1254/10) for the first 24 h. For details of the spheroid setup, see https://doi.org/10.17504/protocols.io.3byl4bnrrvo5/v1. The cells were single-cell dissociated with StemPro Accutase (Thermo Fisher Scientific #A1110501) and seeded at 1000 cells/spheroids in 200 µl Essential 8 Flex supplemented with 10 µM Y-27632 dihydrochloride. The following day, the medium was replenished without Y-27632. Spheroids were processed for experimental perturbations two days following formation.

## Cell line quality control

All cell lines were cultured in the absence of antibiotics except if processed for selection post-engineering as indicated. Cells were routinely tested negative for mycoplasma and genotyped by Sanger sequencing (knock-in lines) or immunoblotting (knock-out lines) to confirm the correct identity prior to experimental use.

## CRISPR/Cas9 gene editing of *PIK3CA* exon 21 in HeLa cells

Low-passage (P5) HeLa cells were used for CRISPR/Cas9 engineering for knock-in of the *PIK3CA* H1047R variant (c.CAT>c.CGT) using a modified version of previously published protocols (Madsen et al, 2019; Madsen and Semple, 2019). Briefly, a total of 200,000 cells were targeted with a total of 200 pmol single-stranded oligodeoxynucleotides (ssODNs) introducing either the targeting mutations along with silent mutations or silent mutations without the targeting mutation: 5'-TAGCCTTAGATAAAACTGAGCAAGAGGCTTTGGAGTATTTCATGAAACAAATGAACGACGCACGTCATGGTGGCTGGACAACAAAAATGGATTGGATCTTCCACACAATTAAACAGCATGCATTGAACTGAAAAGATAACTGAGAAAATG-3' (HDR001 ssODN, with targeting and silent mutations) and 5'-TAGCCTTAGATAAAACTGAGCAAGAGGCTTTGGAGTATTTCATGAAACAAATGAACGACGCACATCATGGTGGCTGGACAACAAAAATGGATTGGATCTTCCACACAATTAAACAG-CATGCATTGAACTGAAAAGATAACTGAGAAAATG-3' (HDR002 ssODN, with silent mutation only). Three different mixtures of ssODNs (HDR001 alone, HDR002 alone, 1:1 mixture HDR001:HDR002) were set up to ensure the generation of a dose-controlled allelic series for *PIK3CA*<sup>H1047R</sup>. Targeting was performed using recombinant ribonucleotide proteins (RNPs) at ratio 1:1.2 (Cas9:sgRNA; 4 µM:4.8 µM). The synthetic sgRNA (5'-AUGAAUGAUGCACAUCAUGG-3') was obtained from Synthego (modified for extra stability). The high-fidelity Alt-R™ S.p. Cas9 Nuclease V3 (IDT #1081061) was used to limit off-targeting risk. Cells were targeted by nucleofection using the SE Cell Line 4D-Nucleofector™ X Kit S (Lonza #V4XC-1032). Cells were allowed to recover from nucleofection before sib-selection-based subcloning to isolate pure clonal cultures. To aid recovery, conditioned medium (1:1 mixture with fresh

medium) was used for 7 days during subcloning. An initial screen for correct genotypes was performed using DNA extracted with QuickExtract (Cambridge Bioscience #QE0905T) and subjected to PCR amplification and Sanger sequencing with primers: 5'-CAGCATGCCAATCTCTTCAT-3' (forward), 5'-ATGCTGTTCATGGATTGTGC-3' (reverse). As HeLa cells are triploid on average, genotypes were called following deconvolution with Synthego's ICE tool (Hsiau et al, 2019). Putative pure WT *PIK3CA* (or silent mutation only) and *PIK3CA*<sup>H1047R</sup> clones were expanded and subjected to final validation by next-generation sequencing using MiSeq, with Illumina adaptor-appending primers: 5'-<u>TCGTCGGCAGCGTCAGATGTGTATAAGAGACAG</u>ATAAAACTGAGCAAGAGGCTTTGGA-3' (forward) and 5'-<u>GTCTCGTGGGCTCGGAGATGTGTATAAGAGACAG</u>ATCGGTCTTTGCCTGCTGAG-3' (reverse). The MiSeq output was analyzed using CRISPResso2 (Clement et al, 2019). All raw and analyzed files are deposited on the accompanying OSF project site (https://doi.org/10.17605/OSF.IO/4F69N, component: *MiSequencing_CRISPR_clone_validation*).

Because HeLa cells are nominally triploid, all knock-in lines with one or two copies of *PIK3CA*<sup>H1047R</sup> harbor two or one allele(s), respectively, with a C-terminal frameshift equivalent to those in the loss-of-function 3xFS clone (Appendix Fig. S1H). This frameshift is too close to the stop codon of p110α to result in nonsense-mediated decay and instead causes a cell clone-dependent change of the last 20–30 C-terminal amino acids of p110α (altered amino acid sequences in the different clones are shown in Appendix Fig. S2B). In all cases, this change abolishes the critical p110α WIF motif required for membrane binding and catalytic function (Jenkins et al, 2023), effectively creating a loss-of-function knock-in that nevertheless remains expressed and thus does not carry the risk of altering the stoichiometry of p85 regulatory and p110 catalytic subunits in these cells.

## Exome sequencing

All CRISPR/Cas9-engineered HeLa clones used in this work were profiled by whole-exome sequencing at an early passage (13 or 14) and compared to the parental HeLa culture prior to editing (passage 4) and following another 10 passages (passage 14) in the absence of editing. This approach enabled evaluation of the extent of mutagenesis caused by the gene editing and single-cloning procedures relative to the expected baseline acquisition of mutations upon prolonged cell culture. High-quality DNA was extracted using the NucleoSpin Micro Kit XS (Takara #740901.5) and submitted to Novogene for exome library preparation and sequencing. Briefly, libraries were prepared with the Next Ultra DNA Library Prep Kit (NEB #E7370L) and enriched for exons using Agilent's SureSelectXT Reagent Kit and Agilent SureSelect Human All ExonV6 (#G9611B). The final libraries were pooled and paired-end (150 bp) sequenced on a NovaSeq 6000 instrument, with 6 G of raw data output per sample. Subsequent read processing was performed with the nf-core/sarek pipeline (v2.6.1), with alignment against the human genome (hg38) and Agilent's reference.bed file corresponding to the SureSelect Human All ExonV6 60MB S07604514 design. Somatic variant calling was performed with Strelka2 (Kim et al, 2018) according to a tumor/normal pairs setup where CRISPR/Cas9-edited clones and the long-term passage parental cultures were assigned the "tumor" label and the low-passage parental culture assigned the "normal" label.

Subsequent variant annotation was performed using the snpEff pipeline (Cingolani et al, 2012). Detailed scripts and multiQC reports for reproducing all nf-core/sarek outputs have been made available on the OSF project site. The snpEff-annotated variants with SomaticEVS filter = PASS were processed with GATK VariantsToTable and imported into R for identification of non-synonymous protein-coding variants that are common to a minimum of two samples when compared to the low-passage parental culture prior to CRISPR/Cas9 gene editing. Intersection plots and heatmaps were generated using the ComplexHeatmap R package (Gu et al, 2016). Clustering was performed according to Euclidean distance with the Ward.D2 method. Raw sequencing data and annotated R processing scripts are provided on the accompanying OSF project site (https://doi.org/10.17605/OSF.IO/4F69N, component: *Exome_sequencing_processed_file_analysis*).

## Total mRNA sequencing

All CRISPR/Cas9-edited HeLa clones were processed for total mRNA sequencing at baseline to determine transcriptional similarities and differences across individual genotypes. Individual clones (passages 16-18) were collected at subconfluence following refeeding with fresh complete medium for 3 h. Following a single wash with DPBS, cells were snap-frozen and stored at −80 °C until further processing. Following thawing on ice, total RNA was extracted using the Direct-zol RNA Miniprep Kit from ZymoResearch (#R2051), with final elution in 30 μl nuclease-free water. Samples were submitted to Novogene for library preparation (NEB Next® Ultra™ RNA Library Prep Kit) and paired-end (150 bp) sequencing on a NovaSeq 6000 instrument. Note that the library preparation is strand-agnostic.

Raw read processing was performed with the Nextflow (version 20.07.1) nf-core RNAseq pipeline (v1.1) (Ewels et al, 2020), with Spliced Transcripts Alignment to a Reference (STAR) (Dobin et al, 2013) for read alignment to the human genome (Homo_sapiens.GRCh38.96.gtf) and featureCounts (Liao et al, 2014) for counting of mapped reads (multimapped reads were discarded). All subsequent data processing was performed in R, with differential gene expression analysis following the limma-voom method (Ritchie et al, 2015). Filtering of low gene expression counts was performed with the TCGAbiolinks package with quantile value 0.75 (chosen empirically based on the observed count distribution). Next, read count normalization was performed with the gene length-corrected trimmed mean of M-values (GeTMM) method (Smid et al, 2018). PCA was done using the PCAtools package. The mean-variance relationship was modeled with voom(), followed by linear modeling and computation of moderated t-statistics using the lmFit() and eBayes() functions in the limma package (Ritchie et al, 2015). The associated p values for the assessment of differential gene expression were adjusted for multiple comparisons with the Benjamini-Hochberg method at false-discovery rate (FDR) = 0.05 (Benjamini and Hochberg, 1995). Adjustments were performed with option = "separate", comparing $PIK3CA^{H1047R}$ mutant clones against WT clones. No differentially expressed genes were identified across the different genotypes at baseline. All R processing scripts to replicate the analyses are provided on the accompanying OSF project site (https://doi.org/10.17605/OSF.IO/4F69N, component: *RNAseq_processing*). The raw sequencing files are available under the GEO under accession number: GSE251956.

## Sleeping Beauty transposon engineering of cells for expression of AKT kinase translocation reporter (KTR)

The Sleeping Beauty transposon-based and optimized miniFOXO kinase translocation reporter (KTR) (Gross et al, 2019) was used to generate stable cell lines from the original CRISPR/Cas9-engineered HeLa cell clones. This was performed at two different locations in two independent sets of WT and mutant clones, with a time gap of one year. For further testing of the reproducibility of the results irrespective of reporter expression levels, stable cell line generation was performed using two different molar ratios of transposon to transposase (the following molar units are for cells seeded in 12-well plates at a density of 50,000 cells/well; these units were scaled by a factor of 2 for cells seeded in 6-well plates at 100,000 cells/well). For a 1:1 molar ratio, engineering was performed with approximately 100 fmol of transposon and SB100X transposase-expressing plasmids (the plasmid maps are deposited on the OSF project site, with code names MB40 and MB43, respectively). The plasmid pSBbi-FoxO1_1R_10A_3D (designated MB40) was a gift from Laura Heiser (Addgene plasmid #106278; RRID:Addgene_106278). The plasmid pCMV(CAT)T7-SB100 (designated MB43) was a gift from Zsuzsanna Izsvak (Addgene plasmid #34879; RRID:Addgene_34879). For a 1:10 molar ratio and thus low reporter expression, engineering was performed with approximately 10 fmol transposon plasmid and 100 fmol transposase plasmid. Plasmid were delivered to cells using Fugene HD Transfection Reagent (Promega #E5912) at a 3:1 Fugene volume:DNA mass ratio for transfection complex formation in Opti-MEM I Reduced Serum Medium (Thermo Fisher Scientific #31985070). Puromycin (Sigma Aldrich #P9620 or #P4512-1MLX10) selection at 1 μg/ml was started 24–48 h after seeding, with replenishment of selection medium at least every second day. Stable cell lines were usually established and banked within 2 weeks of the initial transfection.

## Western blotting

A step-by-step Western blotting protocol is publicly available on protocols.io with the following https://doi.org/10.17504/protocols.io.4r3gv8w. Cells were lysed from 10-cm dishes with RIPA Lysis and Extraction buffer (Thermo Fisher Scientific #89900), and 10–15 μg of protein were loaded on 4–12% Bis-Tris Midi NuPage Protein Gels (Thermo Fisher Scientific) and separated at 120 V for 2 h in NuPage MES running buffer (Thermo Fisher Scientific). Protein transfer was performed with an iBlot2 system (Thermo Fisher Scientific) using program P3. All primary and secondary antibodies used are provided in Appendix Tables S1 and S2. Final signal detection was by enhanced chemiluminescence (ECL) with the Immobilon Forte Western HRP substrate from Sigma Aldrich (#WBLUF0500) or ECL Western Blotting Substrate from Promega (#W1015). Images were acquired on the Amersham ImageQuant 800 system with 5 × 5 binning. All raw Western blots have been deposited on the accompanying OSF project site (https://doi.org/10.17605/OSF.IO/4F69N, component: *Western_blots_Fig.S2*).

## Small-molecule reconstitution and usage

The following growth factors were obtained from Peprotech: human IGF1 (#100-11, lots: 022201-1, 092101-1, 041901-1), human EGF

(#AF-100-15, lots: 0922AFC05, 0222AFC05, 0820AFC05), human Epigen (#100-51, lot: 0706386). Lyophilized stocks were reconstituted in sterile, molecular-grade, non-DEPC-treated water from Ambion (#9937), allowed to dissolve for 15–20 min at 4 °C followed by aliquoting in PCR strip tubes and long-term (up to 1 y) storage at −80 °C. Aliquots were freeze–thawed maximum once to limit loss of potency. Human insulin (10 mg/ml) was from Sigma (#91077 C, lot: 21M018) and stored at 4 °C. BYL719 was obtained from SelleckChem (#S2814, lots: 03, 06) at 10 mM in DMSO. The stock solution was diluted to 1 mM in sterile DMSO, aliquoted in PCR strip tubes and stored at −80 °C long-term (up to 2 years). TGX221 was obtained from MedChemExpress (#HY-10114) and reconstituted at 10 mM in sterile DMSO, prior to long-term storage (up to 3 years) at −80 °C. 1938 was synthesized by Key Organics or SAI Life Sciences and is now available through CancerTools (#161068).

## Phosphoinositide reporter constructs

To minimize confounding effects on reporter performance arising from usage of different plasmid backbone, all PH domain derivatives were cloned into the same plasmid backbone construct (pNES-EGFP-C1 for wild-type PH domains; pNES-mCherry-C1 for mutant PH domains). The generation of each individual reporter is detailed below. Note that all final PH domains are coupled to a nuclear export sequence (NES) and harbor an N-terminal fluorescent protein tag. All plasmids were verified by restriction enzyme digest and Sanger sequencing. Plasmid maps have been deposited on the accompanying OSF project site (https://doi.org/10.17605/OSF.IO/4F69N; component *Other*). All new plasmids generated in the course of this work are available through the MRC Reagents and Services portal https://mrcppureagents.dundee.ac.uk under accession numbers DU78205 (pNES-mCherry-C1-PH-mutARNO-R280A-2G-I303Ex2), DU78206 (pNES-EGFP-C1-PH-mutARNO-R280A-2G-I303Ex2), DU78207 (pNES-EGFP-C1-PH-BTK-WT), DU78208 (pNES-mCherry-C1-PH-BTK-MUT), DU78209 (pNES-EGFP-C1-PH-AKT2-WT), DU78210 (pNES-mCherry-C1-PH-AKT2-MUT), DU78211 (GFP-PH-TH-BTK), DU78212 (GFP-PHmut-TH-BTK).

### ARNO

The pNES-EGFP-C1-PH-ARNO(I303E)x2 PIP$_3$ reporter construct (RRID:Addgene_116868) was a gift from Dr Gerry Hammond (University of Pittsburgh) (Goulden et al, 2019). In this construct, the PH domain is N-terminally tagged with an enhanced GFP (EGFP) which is itself preceded by a nuclear export sequence. To generate a tandem-dimer mutant version equivalent to R280A in the native PH domain of ARNO (Uniprot #P63034), a 996 bp gene fragment corresponding to the tandem-dimer PH-ARNO(I303E) domain with the mutated residues was synthesized as a gene fragment in a pUC vector by GeneWiz, including 5' and 3' HindIII and BamHI recognition sites, respectively. Next, five reactions each with 250 fmol of the construct carrying the mutant fragment or the original WT pNES-EGFP-C1-PH-ARNO(I303E)x2 construct were digested with 20 U each of BamHI-HF (NEB #R3136S) and HindIII-HF (NEB #R3104S), alongside 5 U of quick alkaline phosphatase (calf intestinal, NEB #M0525S), all in 30 μl rCutSmart buffer (NEB) per reaction. The digests were run at 37 °C overnight (16 h), then were heat inactivated at 80°C for 20 min. The digests were then run on a Tris acetate-EDTA agarose gel (1%) and the

pNES-EGFP-C1 destination vector and the mutant PH-ARNO(I303)x2 domain with compatible sticky ends were gel-purified using the Monarch DNA gel extraction kit (NEB #T020S) according to the manufacturer's instructions. The insert and the destination vector were ligated in a 10 μl reaction with 2× instant sticky-end ligase master mix (NEB #M0370S), using a 1:5 molar ratio of backbone-to-insert and otherwise following the manufacturer's instructions. Next, 2 μl of the ligation reaction were heat-shock transformed into high-efficiency 5-alpha competent *E. coli* (NEB #C2987I), followed by conventional colony picking and bacterial culture expansion for subsequent plasmid DNA extraction with the Maxi Plus kit from Qiagen (#12964). Next, the EGFP tag in the new construct containing the mutant PH-ARNO(I303)x2 domain was replaced with an mCherry tag obtained from a pNES-mCherry-C1-TAPP1-cPHx3 construct (RRID:Addgene_116855). The latter was a gift from Dr Gerry Hammond and was originally described in ref. (Goulden et al, 2019). The restriction enzyme digest-based subcloning protocol used for the fluorophore swap was as described above.

### BTK

The PH domain from BTK was obtained from Addgene construct #51463 (RRID:Addgene_51463; a gift from Dr Tamas Balla). This construct was used for site-directed mutagenesis of a key arginine in the PI signature motif (FKKRL) of the BTK PH domain (Cronin et al, 2004) using the following primers: 5'-CTTCAAGAAGgcCCTGTTTCTCTTG-3' (forward) and 5'-TTTAGAGGTGATGTTTTCTTTTTC-3' (reverse). Site-directed mutagenesis was performed with the Q5® Site-Directed Mutagenesis Kit from New England Biolabs (#E0554S), using 0.5 μM of each primer and 0.2 ng/μl plasmid DNA in a 25-μl reaction. The thermocycling conditions were as follows: denaturation at 98 °C for 30 s; 25 cycles of 98 °C for 10 s, 56 °C for 20 s, 72 °C for 2.5 min; final extension at 72 °C for 5 min. The PCR product was subsequently processed for KLD (kinase, ligase, DpnI) treatment as per the manufacturer's instructions.

The wild-type and mutant versions of the BTK PH domain were PCR-amplified, including the addition of 5' and 3' BamHI and HindIII restriction enzyme recognition sites, respectively. The following primers were used (with underlining to indicate the recognition sites): 5'-AGCAGAAGCTTCGATGGCCGCAGTGATTCTGG-3' (forward); 5'-CCGGTGGATCCTCAGTTCTCCAAAATTTGGCAG-3' (reverse primer for PH-TH version, including the addition of stop codon); 5'-CCGGTGGATCCTCACCGGATTACGTTTTTGAGCTGG-3' (reverse primer for PH only version, including addition of stop codon). A two-step PCR amplification was performed using Platinum SuperFi DNA Polymerase (Thermo Fisher Scientific #12351-010) with the following thermocycling conditions: denaturation at 98 °C for 30 s; 5 cycles of 98 °C for 10 s, 60 °C for 10 s, 72 °C for 15 s; 20 cycles of 98 °C for 10 s, 65 °C for 10 s, 72 °C for 15 s; final extension at 72 °C for 5 min. The PCR products were gel-purified and processed for HindIII- and BamHI-based subcloning into the pNES-EGFP-C1 and pNES-mCherry-C1 backbones as described above for ARNO.

### AKT2

The AKT2 PH domain was obtained from a plasmid encoding the full-length AKT2 protein (a gift from Dr James Burchfield (Norris et al, 2017)). This construct was used for site-directed mutagenesis of a key arginine in the PI signature motif (WRPRY) of the AKT2 PH domain (Cronin et al, 2004) using the following primers: 5'-

CTGGAGGCCAgcGTACTTCCTG-3' (forward); 5'-GTCTTGATG TATTCACCAC-3' (reverse). Site-directed mutagenesis was performed as described for BTK except for use of 57 °C as annealing temperature and 4 min of extension time in each cycle. The wild-type and mutant versions of the AKT2 PH domain were PCR-amplified, including the addition of 5' and 3' BamHI and HindIII restriction enzyme recognition sites, respectively. The following primers were used (with underlining to indicate the recognition sites): 5'-AGCAGAAGCTTCGATGAATGAGGTGTCTGTCATC-3' (forward); 5'-CCGGTGGATCCTCAGTTGGCGACCATCTGG A-3' (reverse; this primer also adds a stop codon). The procedure was as described for BTK above, with subsequent subcloning into the pNES-EGFP-C1 and pNES-mCherry-C1 backbones as described for ARNO.

## Live-cell total internal reflection fluorescence (TIRF) microscopy

A detailed protocol of how HeLa cells were prepared for live-cell microscopy by TIRF, including Matrigel coating of the dishes, cell seeding, transfection with phosphoinositide reporters and subsequent treatment has been made publicly available on protocols.io through the following https://doi.org/10.17504/protocols.io.kxygx37jkg8j/v1. The above protocol was also used for experiments with MEFs and A549 with the following modifications. MEFs were seeded at a density of 2000 cells per well (0.35 cm$^2$). A549 cells were seeded at either 2000 or 3000 cells per well and transfected with either 25 ng or 50 ng wild-type and mutant phosphoinositide reporter constructs; these different conditions were tested due to the low transfection efficiency of these cells, however the final results did not differ and were thus pooled together. 20 ng of a pUC19 (NEB #09052008) carrier plasmid was included in all transfection conditions with 25 ng of each reporter plasmid to achieve even uptake and low expression of each PH domain-based reporter, because high expression of PH domain reporters could potentially interfere with endogenous signaling processes.

Time-lapse TIRF images were obtained on a 3i Spinning Disk Confocal microscope fitted with a sCMOS Prime95B (Photometric) sensor for TIRF, with full temperature (37 °C) and CO$_2$ (5%) control throughout the acquisitions. A 100 × 1.45 NA plan-apochromatic oil-immersion TIRF objective was used to deliver the laser illumination beam (488 nm or 561 nm; 40–50% power) at the critical angle for TIRF and for acquisition of the images by epifluorescence (200–300 ms exposure) using single bandpass filters (445/20 nm and 525/30 nm). The acquisition was performed in sequential mode, without binning, using Slidebook 6.0 and an acquisition rate of 70 s.

Image analyses of total reporter intensities were performed with the Fiji open-source image analysis package (Schindelin et al, 2012). The regions of interest (ROI) corresponding to the footprint of the individual cell across time points were defined using minimal intensity projection to select only pixels present across all time points, following prior background subtraction with the rolling ball method (radius = 500 pixels) and xy drift correction. Mean intensity levels for each reporter were measured within the ROI and exported for subsequent data processing in R. All raw images were inspected manually to ensure that measurements were not recorded for time points where a cell underwent apoptosis or migrated out of the field of view. Final trajectory normalizations to the median

signal of pre-stimulus or post-BYL719 time points were performed using the Time Course Inspector package *LOCnormTraj* function (Dobrzynski et al, 2019). The Time Course Inspector package was also used for calculating the mean and bootstrapped confidence intervals of replicate time series data. The image analysis pipeline, including all macros and R analysis scripts used for reporter normalizations and final replicate data processing are provided on the OSF project site (https://doi.org/10.17605/OSF.IO/4F69N, components: *TIRF_analysis_pipeline*, *TIRF_datasets_Figs.1,2*).

## Live-cell epifluorescence microscopy of FOXO-based AKT kinase translocation reporter (KTR)

A detailed protocol of how HeLa cells were prepared for KTR measurements by live-cell widefield microscopy, including Matrigel coating of the dishes, cell seeding, and subsequent treatment has been made publicly available on protocols.io through the following https://doi.org/10.17504/protocols.io.261gedjkjv47/v1. Time-lapse epifluorescence images were obtained on a Nikon Ti2-E Inverted microscope fitted with a high-sensitivity CMOS Prime BSI (Photometric) sensor, with full temperature (37 °C) and CO$_2$ (5%) control throughout the acquisitions. A 10 × 0.45 NA plan-apochromatic dry objective (Nikon) was used for illumination using the following setup: LED-CFP/YFP/mCherry-3X-A Filter Cube; Triple Dichroic 459/526/596; Triple Emitter 475/543/702. Exposure times were 20 ms (for CLOVER) and 50 ms (for mCherry). Acquisition was performed in sequential mode, without binning, using an acquisition rate of 6 min. Nuclear segmentation based on the NLS-mCherry fluorescence intensity was performed with Stardist (Schmidt et al, 2018) in Fiji. Using custom-written Python scripts, the nuclear intensity $KTR_{nuc}$ was calculated as the average intensity of the KTR channel within a 5-by-5 pixel square around the centroid coordinates. Nuclear masks were then expanded by a width of 2 pixels, and the original mask was subtracted from the expanded one to generate the cytoplasmic ring mask. The cytoplasmic KTR intensity $KTR_{cyto}$ was calculated as the average value of the brightest 50% pixels contained within this cytoplasmic ring mask, to avoid inclusion of background pixels in the calculations for cells that were thin and elongated. The nuclear-to-cytoplasmic ratio $CN_R$ was then computed as the ratio of cytoplasmic over total cellular intensities of the KTR sensor:

$$CN_R = \frac{KTR_{cyto}}{KTR_{nuc} + KTR_{cyto}}.$$

The $CN_R$ values were therefore bounded by 0 in the case of pure nuclear intensity (low AKT activity), and 1 in the case of complete nuclear exclusion of the biosensor (high AKT activity).

For trajectory generation, cells were first tracked using Trackmate (Tinevez et al, 2017) in Fiji, based on the centroids generated by the segmentation step. Tracks were filtered by length (only tracks persisting through the full time course were conserved), and (x,y,t) coordinates from cellular tracks were matched with the corresponding $CN_R$ values using custom-written Python scripts. Incorrectly segmented debris are removed from the analysis based on size (<50 pixels) and trajectory (all segmented objects containing <49 time points). Similarly, cell division events are excluded based on trajectory length, and splitting of trajectories not tracked. Multinucleated cells are rare in

the collected datasets, and therefore not separately classified in the analysis. However, due to the nuclei being segmented jointly or separately at random based on the StarDist '2D_versatile_fluo' model, they are typically excluded due to missing trajectories. Cell qualities with any values equal to zero are handled by replacing them with NA in the Python-based pipeline prior to cytoplasmic-to-nuclear ratio calculations. Trajectories to be analyzed presenting with any NA values are omitted by filtering in R prior to final figure generation. Any cells that exhibit saturated intensity values (>4000 based on 12-bit image pixel intensity range) at any time point in the imaging session in the channel used for pixel intensity extraction (CLOVER) are removed from the final analysis in R. Final data analysis in R included visualization and calculation of mean and bootstrapped confidence interval with the Time Course Inspector package (Dobrzynski et al, 2019). All source data and scripts to reproduce the results are deposited on the OSF project site (https://doi.org/10.17605/OSF.IO/4F69N, components *KTR_datasets_-Fig.3,S3, KTR_datasets_2_Fig.3,S3, KTR_analysis_pipeline*).

To minimize bias and to ensure robustness in the final conclusions, KTR experiments were performed independently across a minimum of two different CRISPR clones for each genotype, two different KTR transposon doses, two different experimental sites and three different experimental operators.

## Scaffold-free spheroid generation and experimental processing

Scaffold-free spheroids were generated according to a protocol that is publicly available through the following https://doi.org/10.17504/protocols.io.3byl4bnrrvo5/v1. Spheroids were seeded at 1000-2000 cells/spheroid Corning® Elplasia® ultra-low attachment plates (#4441, #4442) and used for experimentation 48 h later. Prior to growth factor or inhibitor treatments, HeLa spheroids were serum-starved for 4 h by an initial wash in 200 μl (96-well plate) or 3 ml (24-well plate) of DMEM (Thermo Fisher Scientific #41966-029) supplemented with an additional 2 mM of L-Glutamine. Human iPSC spheroids were washed and growth factor-depleted using DMEM/F12 (Thermo Fisher Scientific #21331-046) for 2 h prior to stimulation. In each case, the wash was removed and replaced with 100 μl (96-well plate) or 1 ml (24-well plate) of the same solution. Growth factor and inhibitor solutions were prepared as 3x working solutions and 50 μl (96-well plate) or 500 μl (24-well plate) of each added to the cells when required for a final dilution to 1× (1-100 nM for IGF1 or EGF; 500 nM for BYL719). Corresponding control solutions containing DMSO or non-DEPC-treated sterile water (Ambion #9937) were also applied. At the end of a time course, the spheroids were either processed for multiplexed mass cytometry (96-well plates) or RT-qPCR (24-well plates) as described below. Human iPSC spheroids were processed similarly, except for growth factor removal for 2 hours prior to experimentation, washing with DMEM/F12 (Thermo Fisher Scientific #21331-046) and treatment with 250 nM BYL719.

## Multiplexed mass cytometry (CyTOF) using 3D spheroids

A detailed step-by-step protocol for spheroid fixation, TOB*is* barcoding, enzyme-free single-cell dissociation and subsequent antibody staining for mass cytometry has been made publicly available through protocols.io: https://doi.org/10.17504/

protocols.io.4r3l22bz4l1y/v1. All antibodies used for mass cytometry are listed in Appendix Table S3. Final cell acquisition was performed either on a Helios or an XT mass cytometer, both developed by Fluidigm (now Standard Biotools). Raw mass cytometry data were normalized using bead standards (Finck et al, 2013) and debarcoded as per a previously developed computational algorithm (Zunder et al, 2015). The Mahalanobis and separation cutoff were set to 10 and 0.1, respectively. Debarcoded .fcs files were imported into Cytobank (http://www.cytobank.org/) and gated with Gaussian parameters to remove debris, followed by gating on DNA (Ir-191/193), total S6, RB phosphorylated at Ser$^{807}$/Ser$^{811}$ and CASP3 cleaved at Asp$^{175}$ to separate cell populations according to cell state. For the iPSCs, there were no distinct populations of cells positive and negative for CASP3 cleaved at Asp$^{175}$, which meant that this marker was not used for gating.

Gated populations of interest were exported as untransformed .txt files (excluding header with filename) and pre-processed using the CyGNAL package as previously described (Sufi et al, 2021). The pre-processed .fcs files were imported into R as an object of the *SingleCellExperiment* class and *arcsinh* (inverse hyperbolic sine) transformed with cofactor = 5 using the flowCore (Hahne et al, 2009) and CATALYST (Crowell et al, 2023) packages in R. Individual marker histograms were generated using the ggridges R package. Exact gate settings, debarcoded and gated .fcs files as well as detailed scripts to reproduce all results are available on the OSF project site in dedicated subfolders (https://doi.org/10.17605/OSF.IO/4F69N; component: *Mass_cytometry_CyTOF_-HeLa_Fig.4,S4,S5,S6*). These subfolders also contain exact single-cell numbers for each experimental analysis in plots saved with the file suffix "_total_cell_count_plot.png".

For EMD-PHATE analyses, the pre-processed .fcs files were imported into Python and processed using a custom-written script deposited on the OSF project site. The following markers were used for EMD-PHATE plot generation: "151Eu_pNDRG1 T346", "167Er_pERK1_2_T202_Y204", "173Yb_pS6_S240_S244", "155Gd_pAKT S473", "168Er_pSMAD2_3_S243_S245".

## Reverse transcription-quantitative polymerase chain reaction (RT-qPCR) analysis

Cellular RNA was extracted as described above for total mRNA Sequencing, and 250 ng used for complementary DNA (cDNA) synthesis with Thermo Fisher's High-Capacity cDNA Reverse Transcription Kit (#4368814). Subsequent qPCRs were performed on 2.5 ng total cDNA using PowerUP™ SYBR™ Green Master Mix (Thermo Fisher Scientific #A25742). A fivefold cDNA dilution series was also prepared and used as standard curve for relative quantitation of gene expression. *TBP* was used as normalizer following confirmation that its gene expression was not changing systematically as a function of the tested conditions, which was not the case for *ACTB* (tested as an additional housekeeping gene). Melt curve analyses and separate agarose gel electrophoresis confirmed amplification of the correctly-sized, single product by each primer pair. All primers had amplification efficiencies 95–105%. Samples were loaded in duplicate in 384-well plates. All qPCR data were acquired on a Quant Studio™ 6 Real-Time PCR System (Thermo Fisher Scientific). The thermocycling conditions were as follows (ramp rate 1.6 °C/s for all): 50 °C for 2 min, 95 °C

for 10 min, 40 cycles at 95 °C for 15 s and 60 °C for 1 min, followed by melt curve analysis (95 °C for 15 s, 60 °C for 1 min, and 95 °C for 15 min with ramp rate 0.075 °C/s). All relevant primer sequences are included in Appendix Table S4. Source data and scripts to reproduce the results have been deposited on the OSF project site (https://doi.org/10.17605/OSF.IO/4F69N, component: *RT_qPCR_replicates_combined_Fig.5*).

### Evaluation of cellular morphology

HeLa cells were seeded at a density of 4000 cells/well in black Perkin Elmer ViewPlate-96 dishes (TC-treated, #6005182) coated with Matrigel as per the protocol for TIRF and KTR imaging. After 24 h, 33 µl of 16% methanol-free formaldehyde (Polysciences #18814-20) was added to 100 µl of culture medium in each well to fix the cells in 4% final formaldehyde concentration. Following 15 min of incubation at room temperature away from light, the fixative was removed and cells washed once with 100 µl DPBS, dispensed slowly and at a 45 degrees angle to prevent the cells from dislodging. Next, 75 µl of DPBS was added, followed by 19 µl of 5× fish skin gelatin blocking agent (Biotium #22010) diluted in PBS/T (PBS with 0.05% Tween-20) with 0.5% Triton X-100 (5X concentration, diluted to 0.1% once added to the DPBS). The cells were left to permeabilize and block for 10 min, after which the block/perm solution was removed and replaced with DPBS supplemented with Phalloidin iFlouor 555 (Abcam #ab176756) and HCS CellMaskBlue (Thermo Fisher Scientific #H32720), both diluted 1:1000. Following 30 min incubation at room temperature away from light, the staining solution was removed and the cells washed twice with 100 µl DPBS. Another 100 µl of DPBS was added after the last wash, followed by epifluorescence acquisition on a Nikon Ti2 Eclipse microscope fitted with a 10 × 0.45 NA plan-apochromatic dry objective (Nikon), used for imaging with the following setup: LED-CFP/YFP/mCherry-3X-A Filter Cube; Triple Dichroic 459/526/596; Triple Emitter 475/543/702. Exposure times were 20 ms (for CellMaskBlue) and 60 ms (for Phalloidin and mCherry). The CellMask blue images were converted to jpg for segmentation using Cellpose (v1) (Stringer et al, 2020), and the resulting masks converted to regions of interests (ROIs) using the LabelsToROI plugin in Fiji/ImageJ (Waisman et al, 2021). All edge ROIs were removed. The remaining ROIs were used for calculating the shape properties of the cell masks using the "Measure" function in Fiji/ImageJ. Analysis of variance (ANOVA) models was fit in R to test for differences in solidity and area as a function of genotype. Although the normality assumption was violated for these models, the impact of this is likely to be minimal given the large number of single-cell observations and the assumptions of the central limit theorem. Tukey's Honest Significant Differences method was used to test for statistically significant (adjusted $P$ value < 0.05) differences in shape properties as a function of genotype. All raw images, segmentation masks, quantification scripts and a final montage of all composite images are deposited on the OSF project site (https://doi.org/10.17605/OSF.IO/4F69N, component: *HeLa_cell_morphology_image_analysis_Fig.5*).

### Information-theoretic analyses

For the estimation of mutual information and information capacity, the SLEMI (Statistical Learning-based Estimation of Mutual Information) R package was used (Jetka et al, 2019). SLEMI uses a logistic regression model to learn the discrete probability $P(S|Z)$ of the signal (S) given the response (Z) and subsequently estimates the mutual information, $I(Z;S)$, from the following formula:

$$I(Z;S) = H(S) - H(S|Z)$$

Here, $H(S) = -\sum_s P(s)\log_2(P(s))$ is the entropy of the signal calculated based on the input signal distribution $P(S)$; and $H(S|Z) = E\left[-\sum_s P(s|Z)\log_2(P(s|Z))\right]$ is the conditional entropy of the signal given the response. A uniform signal distribution was used in all cases for mutual information estimation, while information capacity is estimated by maximizing the mutual information over possible signal distributions. This approach does not rely on any form of data binning and is therefore particularly well-suited for the case of high-dimensional outputs such as the time course measurements in live-cell experiments.

The specific scripts used to calculate mutual information are deposited on the OSF project site (https://doi.org/10.17605/OSF.IO/4F69N, components: *TIRF_datasets_Figs.1,2, KTR_datasets_2_Fig.3,S3*).

### Statistics and reproducibility

Data and statistical analyses are detailed in the relevant "Methods" sections. The information-theoretic analyses detailed above also provide the appropriate statistical description of the trajectory datasets as justified by Bayesian decision theory (Voliotis et al, 2014). The mass cytometry datasets are shown as individual distributions, with probability-based thresholding as opposed to use of metrics such as standard deviation and variance given the non-normal distribution of the data. In general, rather than applying conventional statistical tests that would be violated by the structure of our data, we chose to focus on orthogonal validation in independent model systems. Sample size choice reflects best-practice in the field, in addition to empirical evaluation of technical and biological variability specific to each experimental modality. Blinding was not used in this study.

## Data availability

The datasets and computer code produced in this study are available in the following databases. RNA raw sequencing data: Gene Expression Omnibus GSE251956. RNA sequencing count data and analysis scripts: Open Science Framework (https://doi.org/10.17605/OSF.IO/4F69N). Exome sequencing data and analysis scripts: Open Science Framework (https://doi.org/10.17605/OSF.IO/4F69N). MiSequencing raw data and analysis scripts: Open Science Framework (https://doi.org/10.17605/OSF.IO/4F69N). TIRF image analysis data and scripts: Open Science Framework (https://doi.org/10.17605/OSF.IO/4F69N). KTR image analysis data and scripts: Open Science Framework (https://doi.org/10.17605/OSF.IO/4F69N). Information-theoretic analysis scripts: Open Science Framework (https://doi.org/10.17605/OSF.IO/4F69N). Mass cytometry raw data and analysis scripts: Open Science Framework (https://doi.org/10.17605/OSF.IO/4F69N). RT-qPCR data and analysis scripts: Open Science Framework (https://doi.org/10.17605/OSF.IO/4F69N). HeLa morphology data and analysis scripts: Open Science Framework (https://doi.org/10.17605/OSF.IO/4F69N). Raw Western blot data: Open Science Framework (https://doi.org/10.17605/OSF.IO/4F69N). Plasmid maps: Open Science Framework (https://doi.org/10.17605/OSF.IO/4F69N).

The source data of this paper are collected in the following database record: biostudies:S-SCDT-10_1038-S44320-024-00078-x.

## Peer review information

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

## Acknowledgements

The authors would like to thank the following colleagues and collaborators for
technical and scientific advice, including protocol sharing and helpful
discussions: Dr. Gerry Hammond (University of Pittsburgh), Dr. James
Burchfield (University of Sydney), Dr. Alison Kearney (University of Sydney),
Dr. James Opzoomer (UCL Cancer Institute), Dr. Elitza Deltcheva (UCL Cancer
Institute), Dr. Benoit Bilanges (UCL Cancer Institute), Prof. Alex Toker (Beth
Israel Deaconess Medical Centre), Prof. Andre Levchenko (Yale University)
and Prof. Robert Semple (University of Edinburgh). We are indebted to the
following core facilities and their staff for technical assistance throughout this
project: UCL BLIC (Dr. Lucia Conde), UCL Cancer Institute Flow Cytometry
Facility, UCL Cancer Institute Imaging Facility, Dundee Imaging Facility, and
The Francis Crick Institute Flow Cytometry Facility. We are also grateful to
members of the Payne Lab (UCL Cancer Institute) for their assistance with
MiSeq library processing. Finally, we would like to acknowledge the following
funding sources for making this work possible: Wellcome Trust Sir Henry
Wellcome Fellowship 220464/Z/20/Z (RRM), CLOVES Syndrome
Community (RRM), Cancer Research UK grant C23338/A25722 (BV), UCL
Research Capital Infrastructure Fund (BV), UCLH Biomedical Research Centre
grant BRC660a/CAP/DC (BV), Cancer Research UK grant C60693/A23783
(CJT), Cancer Research UK City of London Centre grant C7893/A26233 (CJT),
UCLH Biomedical Research Centre grant BRC422 (CJT), Cancer Research UK
core funding grant CC2040 (ES and ALM), UK Medical Research Council core
funding grant CC2040 (ES and ALM), Wellcome Trust core funding grant
CC2040 (ES and ALM), CAN_ORGANISE ERC Advanced Grant 101019366 (ES
and ALM), AstraZeneca (ALM), RESETageing H2020 grant 952266 (VIK),
VitaDAO/Molecule academic partnership (VIK), Longaevus Technologies
grant (VIK), Lilly Research Award 28008 (VIK).

## Author contributions

**Ralitsa R Madsen**: Conceptualization; Data curation; Software; Formal analysis;
Supervision; Funding acquisition; Validation; Investigation; Visualization;
Methodology; Writing—original draft; Project administration; Writing—review
and editing. **Alix Le Marois**: Data curation; Software; Formal analysis;
Validation; Investigation; Writing—review and editing. **Oliwia N Mruk**: Data
curation; Software; Formal analysis; Validation; Investigation; Visualization;
Writing—review and editing. **Margaritis Voliotis**: Formal analysis; Writing—
review and editing. **Shaozhen Yin**: Validation; Investigation. **Jahangir Sufi**:
Resources. **Xiao Qin**: Resources; Writing—review and editing. **Salome J Zhao**:
Investigation. **Julia Gorczynska**: Investigation. **Daniele Morelli**: Resources.
**Lindsay Davidson**: Resources. **Erik Sahai**: Resources; Funding acquisition.
**Viktor I Korolchuk**: Supervision; Writing—review and editing. **Christopher J
Tape**: Resources; Supervision; Funding acquisition. **Bart Vanhaesebroeck**:
Supervision; Funding acquisition; Writing—review and editing.

Source data underlying figure panels in this paper may have individual
authorship assigned. Where available, figure panel/source data authorship is
listed in the following database record: biostudies:S-SCDT-10_1038-S44320-
024-00078-x.

## Disclosure and competing interests statement

RRM has received consulting fees from Nested Therapeutics (Cambridge, U.S.)
and serves on the Scientific Advisory Board of CLOVES Syndrome Community.
BV is a consultant for iOnctura (Geneva, Switzerland) and Pharming (Leiden,
the Netherlands) and a shareholder of Open Orphan (Dublin, Ireland). ES is a
consultant for Phenomic AI (Toronto, Canada) and Theolytics (Oxford, UK),
receives research funding from AstraZeneca, MSD and Novartis. VIK is a
scientific advisor for Longaevus Technologies. The remaining authors declare
no competing interests.

# Expanded View Figures

**Figure EV1.   Systematic benchmarking of pleckstrin homology (PH) domain-based class I PI3K biosensors.**                                                    ▶

(**A**) Schematic of the optimized live-cell imaging setup to ensure that multiple comparisons could be performed in the same microenvironment, aided by fluidics for minimal physical perturbation during compound additions. To bring down the baseline of PI3K signaling, serum was removed from the cells 3 h prior to imaging start. D1, D2, D3 refer to day 1, day 2 and day 3 of the experimental workflow. (**B**) Schematic of the different wild-type and mutant PH domain constructs used for benchmarking, all cloned into the same plasmid backbone for consistent comparisons. The portion of the $PIP_3$-binding region in the PH domain of GRP1, which has often been used for live-cell detection of $PIP_3$, has an identical sequence to that in the ARF GEF ARNO. We therefore chose to include the latter in our comparisons given the tandem-dimer, modified version of this PH domain as biosensor for $PIP_3$ (Goulden et al, 2019).The shown alignments cover the conserved β1 strand, variable loop 1, and β2 strand of the PH domain fold. Of the four PH domains, only PH-AKT2 is capable of binding both $PIP_3$ and $PI(3,4)P_2$. The remaining PH domains only bind $PIP_3$ (Posor Y et al, 2022). The alanine (A) mutation in the phosphoinositide (PI) signature motif renders the mCherry-tagged mutant PH domain versions unable to bind phosphoinositides. (**C**) Quantification of total internal reflection fluorescence (TIRF) microscopy experiments comparing the response rate and dynamic range of individual PH domain-based PI3K reporters in response to pharmacological PI3Kα activation in HeLa cells. To correct for non-specific increases in biosensor signal at the plasma membrane, the intensity of each GFP-tagged wild-type PH domain was normalized to that of its mCherry-tagged mutant version. Experimental replicates and single-cell numbers are indicated. Two different configurations were tested for the BTK-derived PH domain: with and without the adjacent Tec homology (TH) domain. Only one experiment was performed with PH-BTK without TH because most of the cells failed to tolerate its expression. (**D**) TIRF microscopy of the PH-AKT2-derived biosensor in HeLa cells stimulated with 5 µM 1938, then treated with the PI3Kα inhibitor BYL719 (500 nM). Two independent experiments are superimposed to illustrate the expected inter-experimental variability. (**E**) Evaluation of the performance of the PH-TH version of BTK with N-terminal or C-terminal fluorescent protein fusion, with simultaneous removal of the nuclear export sequence, as in the original plasmid DNA used for subcloning of this reporter. All plots in (**C**, **D**, **E**) represent mean normalized reporter signal relative to time 0, with shading corresponding to $+/-$ 1 standard deviation (SD).

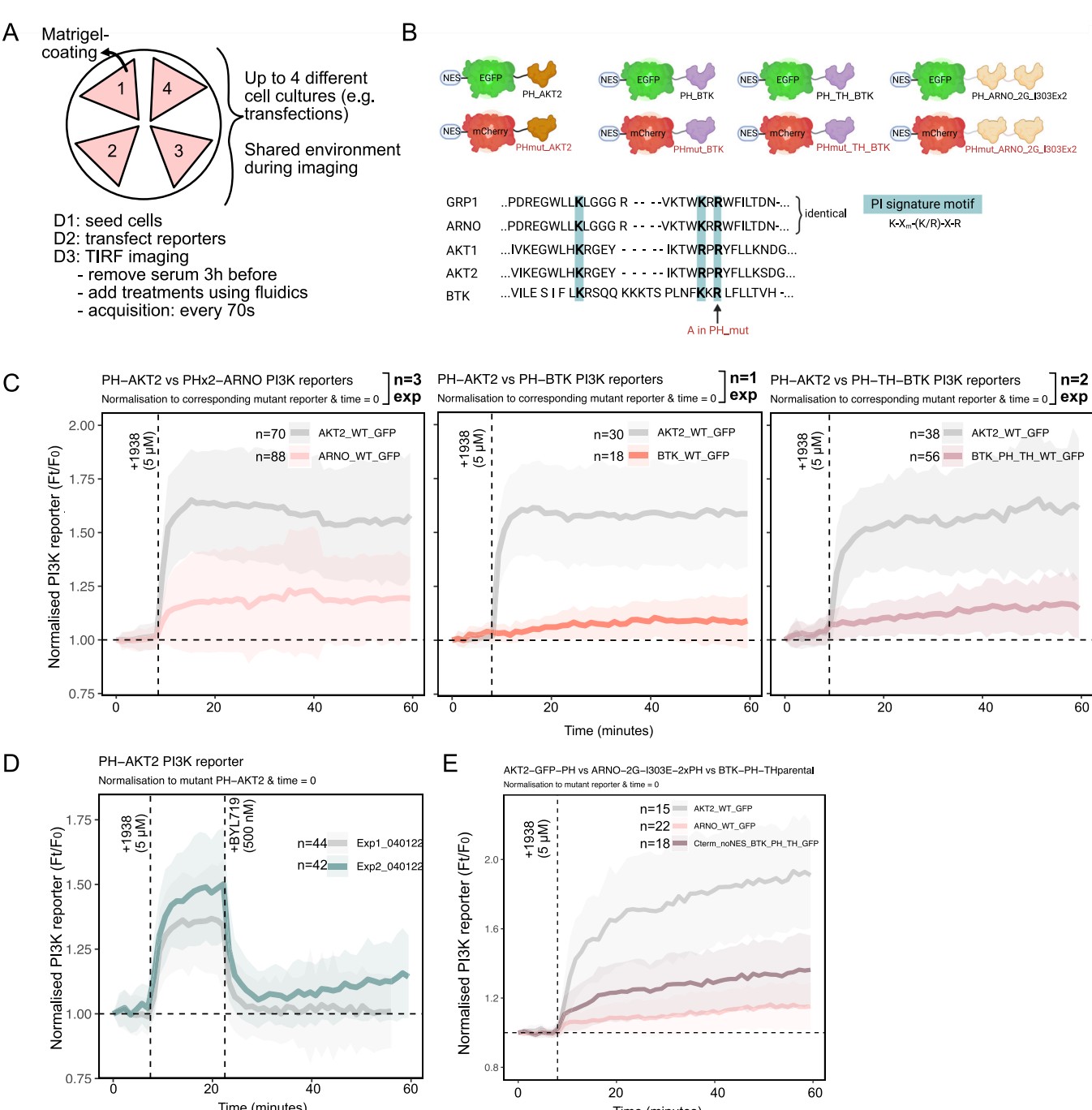

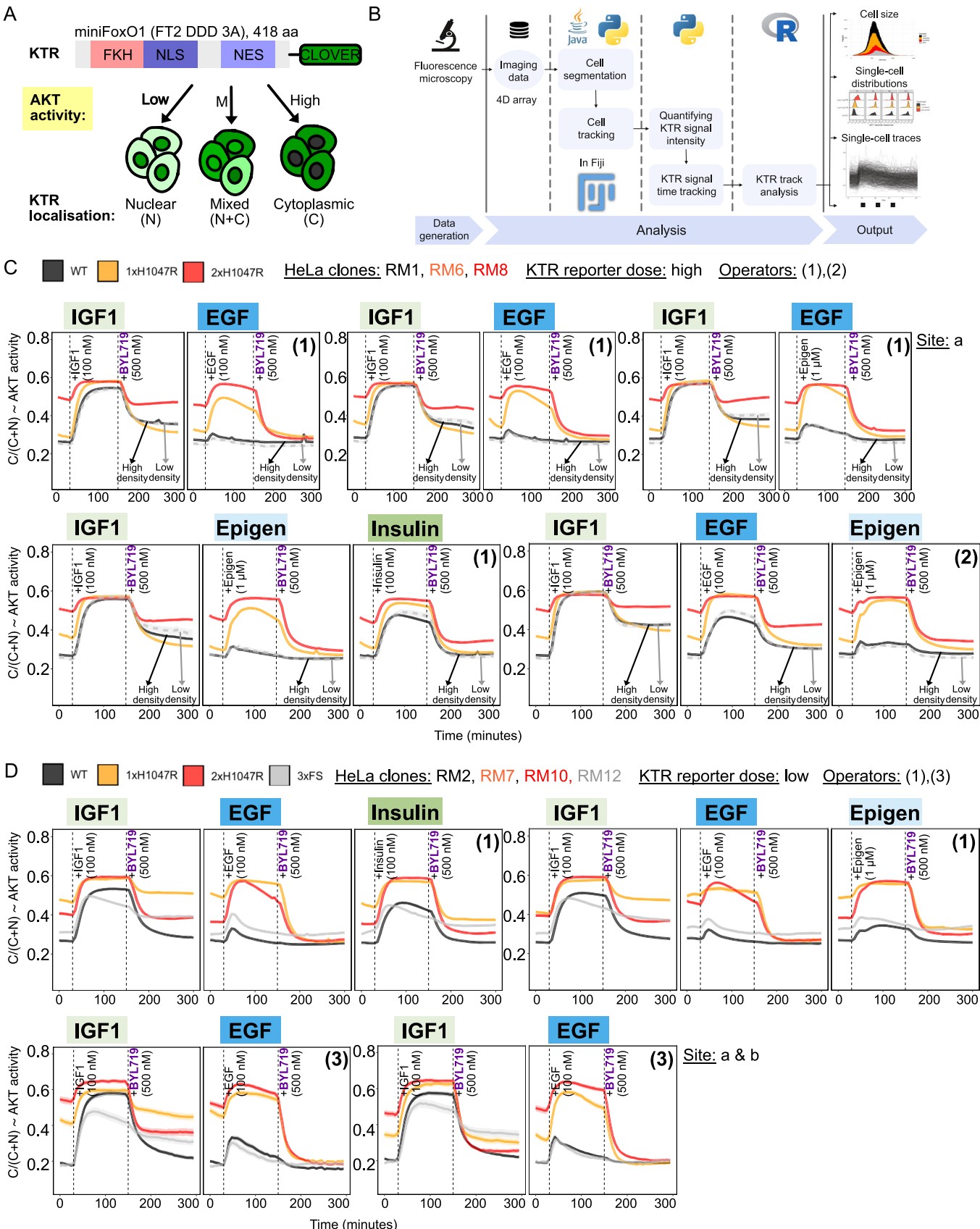

◄    **Figure EV2.   FOXO-based AKT kinase translocation reporter (KTR) setup and full set of experimental outputs.**

(**A**) Schematic of the reporter, which was expressed stably in cells using transposon-based technology, and its mechanism of action. (**B**) Overview of the computational image and KTR data analysis pipeline which has been deposited on the accompanying OSF project site (10.17605/OSF.IO/4F69N). (**C, D**) The data in (**C, D**) are from all independent experiments performed across different genotypes, HeLa clones, cell densities, KTR reporter doses, operators and experimental sites for a robust evaluation of reproducibility. Experiments with high and low KTR transposon dose are shown in (**C**) and (**D**), respectively. For each time point, the traces correspond to the mean proportion of cytoplasmic KTR signal, with shaded areas representing bootstrapped 95% confidence intervals of the mean (note that these may be too small to be seen on the figure). Although we observed operator-dependent differences in EGF-induced signaling dynamics in WT *PIK3CA* cells, the overall pattern relative to IGF1, including the blurring of the response in mutant cells, remained consistent. This technical variability in EGF responses in WT cells is likely due to their sensitivity to the pressure/rate of delivery of the stimulus through the manual fluidics system (see https://doi.org/10.17504/protocols.io.261gedjkjv47/v1).

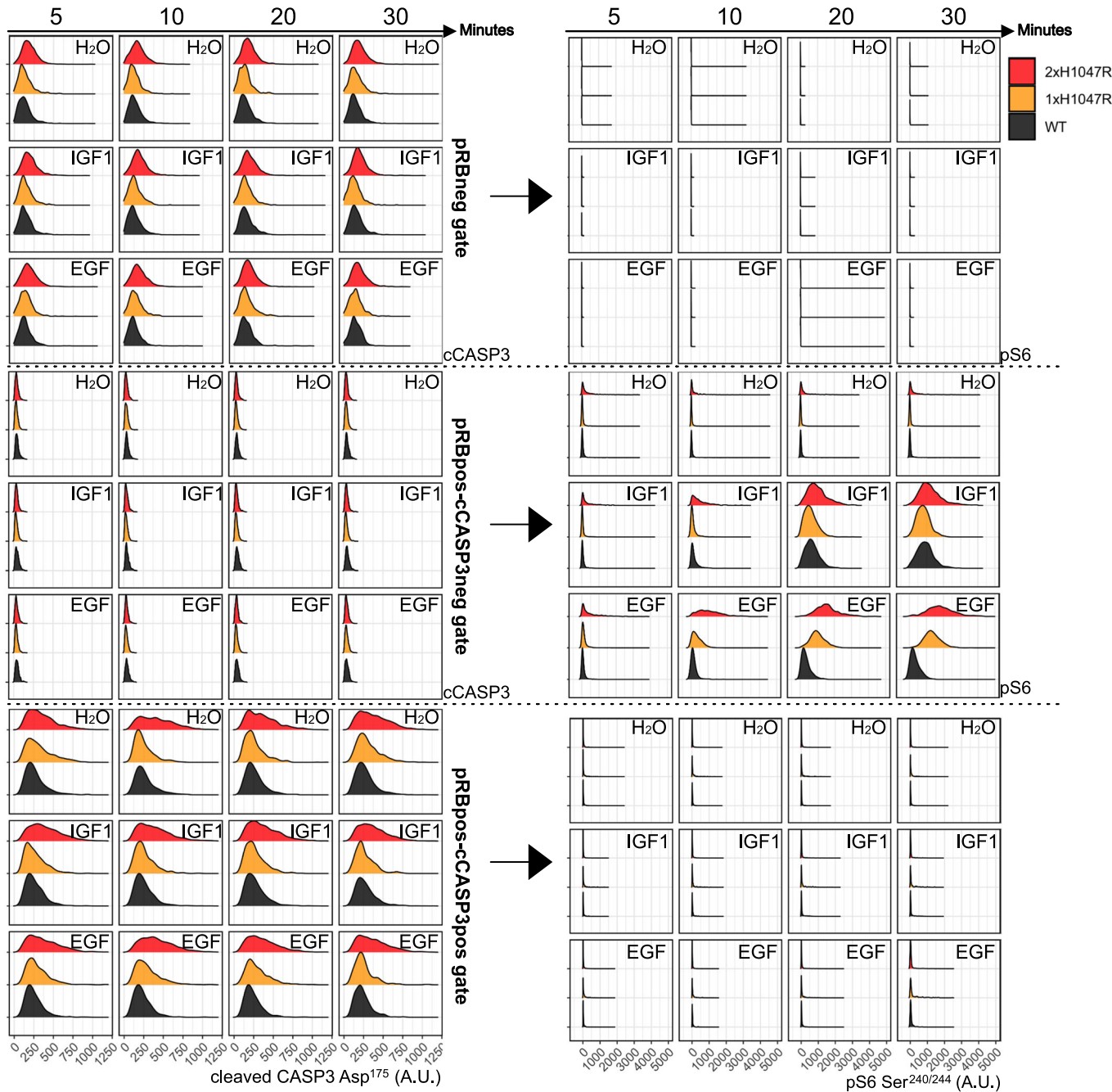

**Figure EV3. Representative CyTOF data demonstrating that growth factor-induced signaling responses are observed only in cycling and non-apoptotic HeLa spheroid cells.**

The plots on the left-hand side show the single-cell signal for cascade 3 (CASP3) cleaved at $Asp^{175}$ in the different pRB gates (pRB-negative$^-$ or pRB-positive$^+$ at $Ser^{807/811}$). The plots on the right show the corresponding pS6 $Ser^{240/244}$ signal in each gate. The overall experimental setup is as shown in Fig. 4. The shown data are from $n = 1$ clone per genotype but are representative of four independent experiments across two independent clones per genotype.

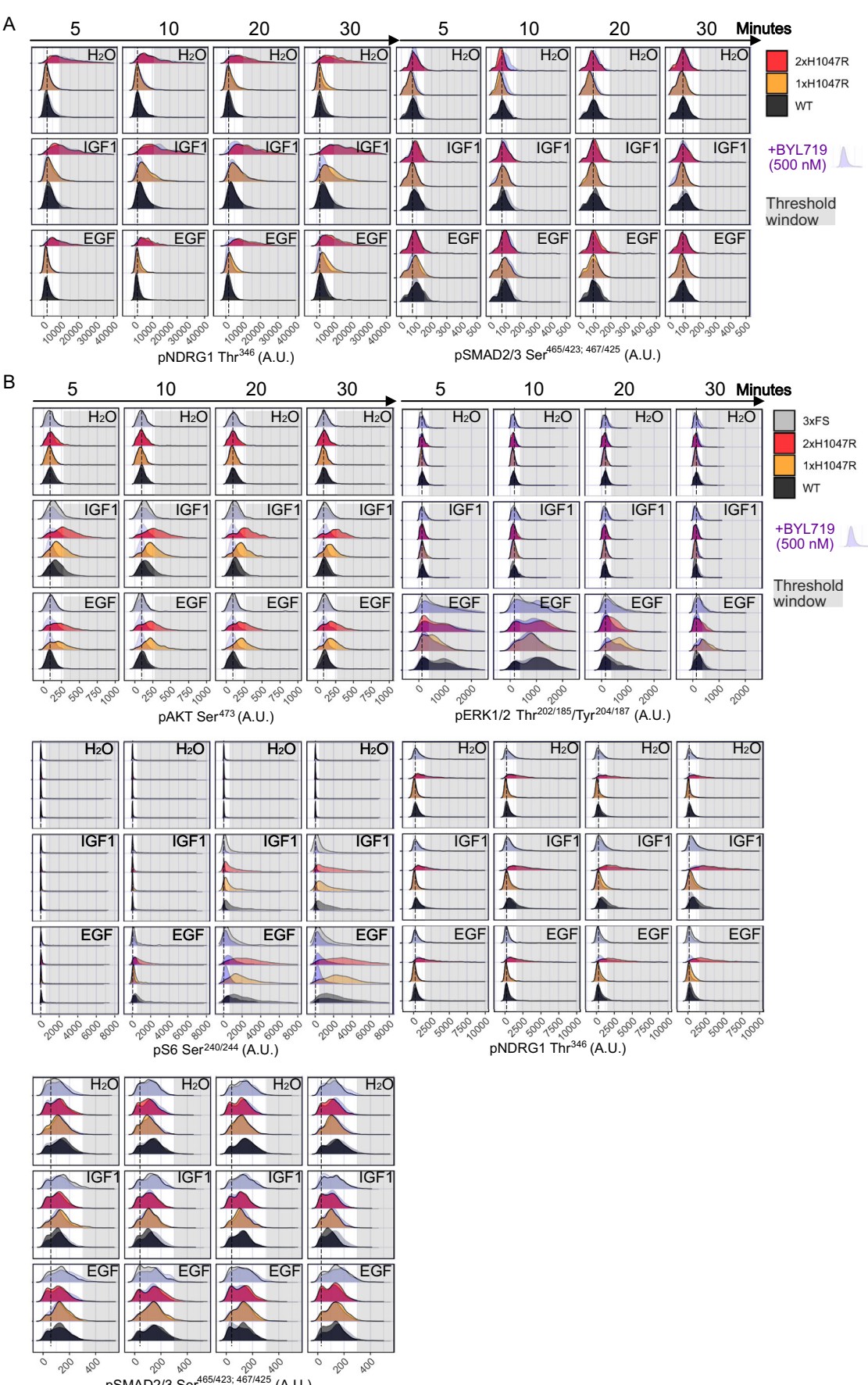

**Figure EV4.  Additional CyTOF data and independent experimental replicate with independent clones.**

(A) The pNDRG1 and pSMAD2/3 signaling responses captured as part of the dataset shown in Fig. 4. The phosphorylation of NDRG1 on $Thr^{346}$ is a marker of mTORC2 activation (García-Martínez and Alessi, 2008). The phosphorylation of SMAD2/3 ($Ser^{465/423}$; $Ser^{467/425}$) is a marker for activated TGFβ signaling which is associated with $PIK3CA^{H1047R}$ phenotypes in human iPSCs (Madsen et al, 2021). (B) CyTOF data from an independent repeat of the experiment in Fig. 4, using independent CRISPR/Cas9-engineered, 3D-cultured HeLa clones, including the *PIK3CA* loss-of-function 3xFS clone as an additional control. The spheroids were serum-starved for 4 h prior to the indicated perturbations. The signaling data are from cycling, non-apoptotic cells. The stippled line indicates the position of the peak in WT spheroids treated with vehicle ($H_2O$). The gray shading highlights the response region not shown by WT *PIK3CA*-expressing cells in the absence of stimulation.

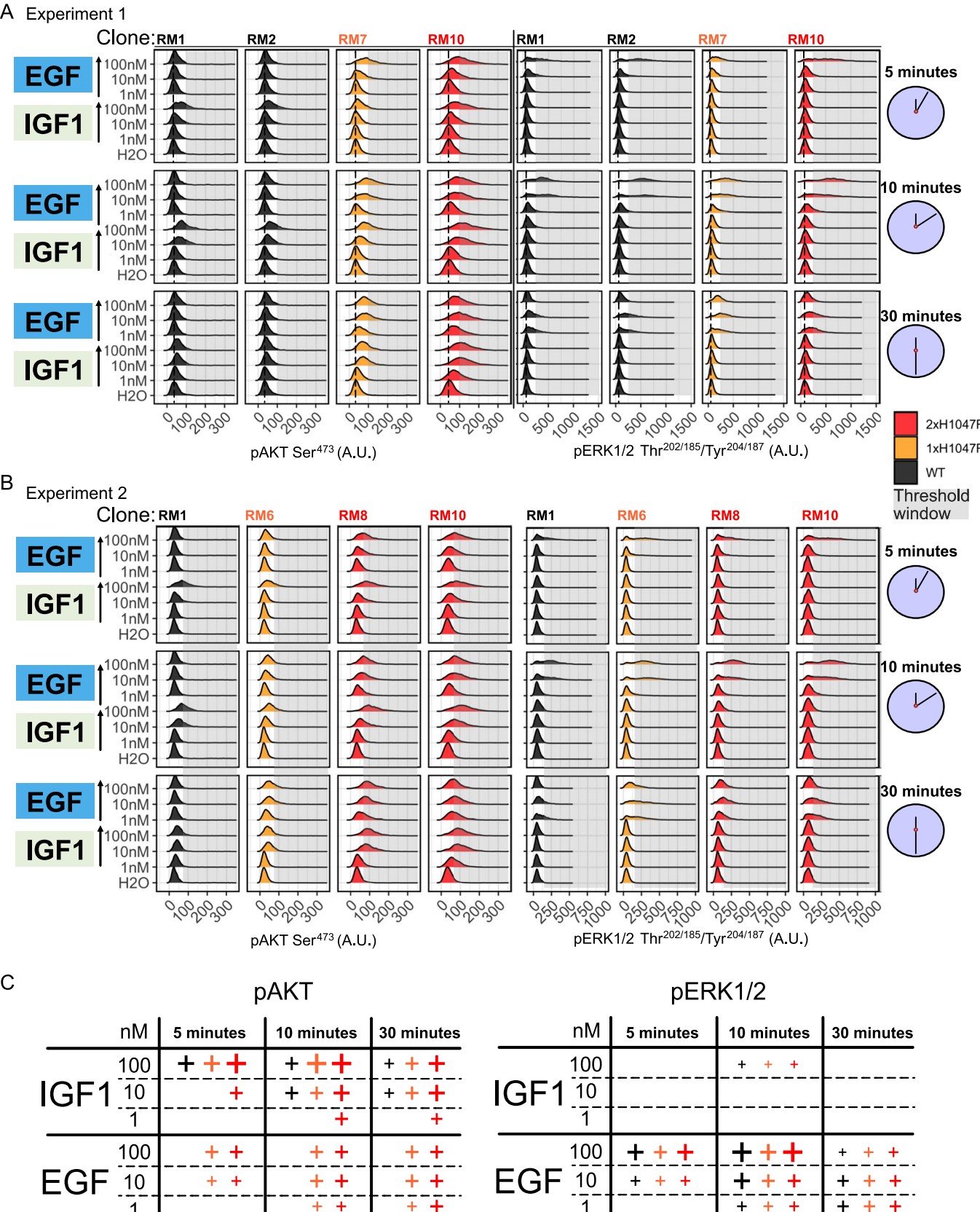

◀ **Figure EV5. Dose- and time-dependent IGF1 and EGF single-cell signaling responses in HeLa spheroid cells with WT or *PIK3CA*$^{H1047R}$ (1–2 copies) expression.**

(A, B) The plots in (A, B) are from two independent CyTOF datasets using independent CRISPR/Cas9-engineered, 3D-cultured HeLa clones stimulated with 1, 10 or 100 nM of IGF1 or EGF as a function of time. The spheroids were serum-starved for 4 h prior to the indicated perturbations. The signaling data are from cycling, non-apoptotic cells. The stippled line indicates the position of the peak in WT spheroids treated with vehicle (H$_2$O). The gray shading highlights the response region not shown by WT *PIK3CA*-expressing cells in the absence of stimulation. (C) Graphical summary of the key observations in the datasets in (A, B). A positive response is indicated with (+), the size of which indicates the response magnitude.

