## [Peer Review File · Molecular Systems Biology]

Oncogenic PIK3CA corrupts growth factor signaling specificity

Ralitsa Radostinova Madsen, Alix Le Marois, Oliwia Mruk, Margaritis Voliotis, Shaozhen Yin, Jahangir Sufi, Xiao Qin, Salome Zhao, Julia Gorczynska, Daniele Morelli, Lindsay Davidson, Erik Sahai, Viktor Korolchuk, Chris Tape, and Bart Vanhaesebroeck

Corresponding author(s): Ralitsa Radostinova Madsen (RMadsen001@dundee.ac.uk)

Review Timeline:	Submission Date:	21st Oct 24
	Editorial Decision:	31st Oct 24
	Revision Received:	6th Nov 24
	Accepted:	11th Nov 24

Editor: Poonam Bheda

Transaction Report: The first review round of this manuscript was performed in another journal.

Reviewer #1

Madsen et al. observe that oncogenic PIK3CA corrupts growth factor signaling specificity. This is a vastly important cancer signaling-related observation. In support of this, the authors present optimized single-cell-based kinetic workflows that they constructed for systematic mapping of quantitative signaling specificity in the PI3K/AKT pathway, and a framework for precise calculations of PI3K-specific information transfer for different growth factors. This remarkable effort produces a much-needed framework assessing how the expression of PIK3CA^{H1047R} corrupts growth factor-specific signaling, altering cell fate.

The authors observe amplification of EGF-induced responses and increased phenotypic heterogeneity in an allele dose-dependent manner, in line with an earlier (2009) knockin of mutant PIK3CA activation report. While the trend was expected, this work provides a quantitative model of growth factor-specific mechanisms altered by an oncogenic mutation. As cell biology marches toward quantification, this work provides a prime, signaling-connected example. This is particularly important in this case, as the study focuses on a highly mutated protein in cancer in the PI3K/AKT/mTOR pathway, with critical crosstalk and feedback loop rewiring with MAPK (ERK). It may also permit construction of kinetic workflows of other combinations of signal-stimulating events.

The experiment-based mechanistic model observes that the mutant-stimulated signaling is not simply an ON switch of the PI3K/AKT pathway. Other factors are involved as discussed in the references pointed out by the authors (8,31). Both references (including PMID: 35121384) emphasize that quantitation is vital, but to date not taken up. In the case of the mutation amplifying the EGF-stimulated signal, the authors suggest that the enhancement by the PIK3CA^{H1047R} signaling is due to the longer residence time at the membrane, which is a reasonable explanation considering the mode of action of this mutation.

Their quantitative single-cell observations of PI3K/AKT signaling dynamics indicate that PIK3CA^{H1047R} expression increases signaling and phenotypic heterogeneity. Heterogeneity is a hallmark of cell populations, reflecting the cell lineage and history. The increased heterogeneity observed with higher signaling levels due to the mutation is expected and may point to activation of multiple oncogenic pathways involving altered networks, including feedback loops/crosstalk in the different cell states such as the MAPK/ERK phosphorylation. Increased activation of EGFR signaling by overexpressed, mutation-free PI3K could also lead to mTOR activation, and increased heterogeneity through multi-pathway signaling.

This comprehensive work stands out with its deep understanding, and insight. It probes several mitogens (IGF1, insulin, EGF, epigen), and did not deem it enough to e.g., simply observe the average trajectories to determine whether the FOXO-based signaling dynamics are sufficiently distinct to allow

individual growth factor inputs to be differentiated from one another. They also did not stop upon observing that the EGF-induced PIP3/ PI(3,4)P2 reporter response in mutant cells appeared amplified and largely indistinguishable from that of IGF1 in wild-type cells. Instead, it led them to further hypothesize that PIK3CA^{H1047R} expression may corrupt the cell's ability to resolve different growth factor inputs from one another. For this to have any significance, however, it would need to be reflected in the activity of key effectors downstream of PIP3/PI(3,4)P2 generation, which they did probe.

This work may trigger a drive to quantify productive signaling and the parameters that define it. Determination of signal propagation can point to cell proliferation, emerging drug resistance and identification of new therapeutic targets. To date, signaling has been described phenomenologically. The common description of signal transduction as propagating from one node to another down the pathway falls short. It is unable to provide in-depth understanding of this vital process in life.

Theory has long postulated that single mutations are insufficient to cause cancer, and multiple publications addressed the question of the minimal number that is required. Would double, or multiple mutations increase of the signaling level and the heterogeneity of the population? If signaling is too strong, the outcome can be irreversible senescence (OIS). But if at that level, considering this model, what would the authors expect?

AUTHORS' RESPONSE: *We thank the Reviewer for their time and in-depth appreciation of the importance of our work, including the delivery of a “much-needed framework” for assessing oncogenic PI3K signalling in quantitative terms. Our intention with this work is indeed to provide the community with rigorous and optimised protocols that allow quantification of “productive signaling and the parameters that define it”. As noted by the Reviewer, this will contribute to ongoing efforts that seek to advance signalling from purely phenomenological, reductionist-centric descriptions towards in-depth quantitative and predictive understanding.*

Regarding the Reviewer's question about the impact on heterogeneity of double/multiple mutations, our data provide insights when it comes to multiple copies of the same mutation. We do indeed see increased PI3K/AKT and MAPK/ERK signalling amplitude and heterogeneity (for PI3K/AKT) in the presence of 2x H1047R copies compared to 1x H1047R – both in cervical cancer cells (HeLa) and in human induced pluripotent stem cells. We are cautious not to extrapolate to the case of multiple different PIK3CA mutations without direct testing. Other mutations, for example those in the helical domain of PIK3CA have different mechanisms of action, and will thus require evaluation in their own right to determine if they, too, increase signalling heterogeneity and loss of specificity. This is now possible to do at scale with our new PI3K-focused single-cell signalling framework and is indeed a planned follow-up to this work.

As to the question about phenotypic consequences of multiple mutations, including senescence and/or cancer development, we have made the following observations

that may be of interest. Despite a very efficient CRISPR/Cas9 knock-in workflow, we never obtained HeLa cells with the mutations knocked into all three *PIK3CA* alleles, yet we saw a substantial enrichment for clones with 1x and 2x *PIK3CA*^{H1047R} knock-in copies (~38% of all expanded clones, n=21). We even observed successful knock-in of a silent mutation across all three *PIK3CA* alleles (clone RM1 in the manuscript). This suggests that 3x *PIK3CA*^{H1047R} copies would be toxic to HeLa cells, potentially due to senescence as we do observe a higher percentage of larger, senescent-like and multinucleated cells in 2x *PIK3CA*^{H1047R} HeLa clones. This observation is in line with the general thinking in the field, including recent work demonstrating high sensitivity of cancer cells to excess oncogenic signalling activity (Dias et al. 2024). In terms of progression from benign to malignant disease, our prior work with *PIK3CA*-related overgrowth spectrum (PROS) disorders and human iPSC models thereof does indeed conclude that heterozygosity for *PIK3CA*^{H1047R} is not sufficient to cause cancer. For more in-depth information, the Reviewer may find Ref. 5 in the original manuscript of interest. It is a review of ours and includes information from mouse models of endogenous *PIK3CA*^{H1047R} expression, once again supporting the notion that heterozygosity for *PIK3CA*^{H1047R} is not enough for cancer to develop.

This study utilizes a GFP-AKT2-PH domain construct as a fluorescence microscopy reporter of PI3K lipid products in HeLa cells, A549 cells and MEFs to determine the relative activity of PI3K under basal and acute growth factor-stimulated states, focused on IGF1 and EGF. PI3Ka loss, via genetic ablation or BYL719 treatment, eliminated between 40 and 60% of the response, indicating redundant contributions from other PI3K isoforms. The vast majority of this study is limited to single HeLa cell clones expressing 0, 1, or 2 copies of oncogenic PI3Ka. Unsurprisingly, it is found that oncogenic PI3Ka increases basal PI3K activity and also enhances the responses to IGF1 and, even more so, to EGF in HeLa cells. However, dependence of this activity on EGF dose appears to be altered by oncogenic PI3Ka in HeLa cells. Oncogenic PI3Ka expression also increases the variability between cells of signaling responses downstream of IGF1 and EGF in cultured HeLa cell spheroids. Somewhat similar effects were seen in iPSCs with two alleles of oncogenic PI3Ka. The more robust signaling induced by EGF in oncogenic PI3Ka HeLa cells was also reflected in enhanced induction of downstream transcriptional responses and correlated with morphological changes.

There are some interesting and potentially novel observations made in this study. However, potential mechanisms underlying these observations are not determined or even tested.

AUTHORS' RESPONSE: We thank the Reviewer for taking the time to assess our work and for their thoughtful comments. We decided to focus here on systematic mapping of the quantitative flow of biochemical information in the PI3K pathway and its corruption by defined, disease-related perturbations. We do this using novel workflows and certainly regard this is “mechanistic” - and complementary to the reductionist-based investigations of signalling pathway “anatomy” that are common in the field, and to which we think the reviewer refers. We build on prior knowledge of molecular mechanisms and, crucially, provide direct mechanistic evidence that oncogenic PIK3CA acts by corrupting dynamic signal encoding, along with increased single-cell heterogeneity. This is in contrast to the conventional yet inaccurate view of PIK3CA^{H1047R} as a simple ON switch of the pathway. We moreover expect our findings and detailed workflows to be widely applicable in attempts to understand and model the origins of incomplete phenotypic penetrance in human diseases of aberrant PI3K activation. This fills a key methodological gap in the field. Our work has the potential to unify disparate findings from independent PI3K signalling studies by enabling quantitative evaluation of cell-specific phenotypic thresholds and signal-specific response variability (see also Refs. 7, 8 in the original manuscript). This is a crucial contribution of our work. For a more in-depth discussion of “stochasticity” as a mechanism of human disease, we recommend the following reviews: Levchenko 2023; Feinberg and Levchenko 2023; Jenkins 2024).

Specific Comments:

1. The authors use the term “blurred biochemical vision” to explain the effects of oncogenic PI3Ka on the EGF response in HeLa cells. The cells are basically going from low EGF responsiveness to very high responsiveness for PI3K activation, which is more comparable to that documented for IGF1 and Insulin in HeLa cells. Again, no mechanism is tested or provided to explain the changes that are reported, although many known factors could influence this response (e.g., changes in EGFR levels or plasma membrane retention, expression of ErbB3 (which unlike EGFR, directly activates PI3K), changes in the expression of specific scaffolding adaptors, etc). It is also not entirely clear whether such changes are biological or the result of clonal variation in the edited HeLa cells.

AUTHORS’ RESPONSE: *We have preliminary evidence that the mechanism underlying the observed “blurring” reflects a PIK3CA^{H1047R} / GAB1 / RAS positive feedback loop. We allude to this in the discussion of the current manuscript, however including all additional mechanistic work here will dilute the current manuscript and its core focus on quantitative signalling. As our current follow-up datasets include mapping of the wider PIK3CA^{H1047R} interactome, we agreed with the Science editor that such follow-up is better reserved for an independent study in its own right. We therefore look forward to sharing these results in due course once all additional data – including mechanistic validations – have been performed. The current manuscript will, however, be of broad interest to the readership of Science Signaling and will thus benefit from timely publication.*

Regarding the Reviewers concern about clonal variability in HeLa cells, we note that we validate key findings in a completely independent cellular model system – human iPSCs which also feature allele dose-dependent endogenous expression of PIK3CA^{H1047R}. Both HeLa and the iPSC systems feature multiple independent clones and benefit from extensive validation (published previously for the iPSCs). In, Figure S2, we provide exome sequencing, RNA sequencing and basic Western blotting QC of all HeLa clones used in this study. In short, the consistency in the observed loss of signalling fidelity across independent PIK3CA^{H1047R} clones cannot be explained by clonal variability or off-target effects.

2. Similarly, the mechanisms underlying the described variable response to IGF1 and EGF in HeLa cell spheroids and, to a lesser extent, in iPSCs with oncogenic PI3Ka are not explored.

AUTHORS’ RESPONSE: *We are currently investigating this as part of a recently funded PhD project. Briefly, for the Reviewer’s interest, the mechanism appears to be related to mechanical tension which varies across the cell population as a function of local density differences. We are dissecting the exact molecular interactions to exploit this therapeutically. The results will be reported in an independent study to ensure that we have the time and space to perform and report all experiments comprehensively. As per the above, we agreed with the Science editor that this would be outside the scope of the current manuscript, and take longer to deliver.*

Madsen et al. analyze the effects of an oncogenic mutation in PIK3CA (H1047R) on PI3K signaling in response to ligands that activate insulin receptor and EGFR. They use a pleckstrin homology domain sensor to monitor PI3K product production and spend the first part of the paper evaluating these sensors before settling on the PH domain of AKT2. The authors then assess PI3K product formation in response to boluses of IGF1 and EGF at 3 different concentrations in HeLa cells and argue that mutant PIK3CA erodes signaling fidelity in a growth factor specific manner. They then analyse AKT translocation to the nucleus and conclude from the data that cells with mutant PIK3CA lose ability to distinguish insulin/IGF1 ligands from EGF ligands – blurring biochemical vision as they put it. Next, the authors turn to mass cytometry studies in spheroids, assessing AKT, ERK, and S6 phosphorylation (and others). In these data, the PIK3CA mutations seem to influence responses to IGF1 and EGF as expected, but ERK phosphorylation looks quite different for the two ligands. They do look more similar for AKT, though, as seen in Figure 3. The authors interpret their PHATE analysis as suggesting that there is an overall “blurring” of signaling response. Note that HeLas with 1x vs 2X of the mutant are quite different, so the degree of blurring depends on type and CN of mutant (which makes the concept tricky). The authors also argue that responses are more variable with the PIK3CA mutants. This seems to be more true for AKT than ERK, though, and probably more in Figure 6 than 4. Finally, they assess IEGs and DEGs in the HeLa spheroids, and find that responses characteristic of EGF were increased by PIK3CA mutant, including SNAIL. This is interesting, but its origin is not clearly explained. Overall, there are some interesting observations and approaches here, and the results suggest that expressing PIK3CA mutants does corrupt signaling. It is not clear to this reviewer that the analysis presented here provides sufficient new insight on this to warrant publication in Science. The analysis is quite sophisticated, but the benefits of the sophistication of the analysis are not immediately clear. The interpretations remain quite phenomenological, with apparent blurring, increased heterogeneity, and elevated background being central. Much of this is evident from visual inspection of the data, and the more detailed analysis does not seem to deepen mechanistic insight very much.

AUTHORS’ OVERALL RESPONSE: *We thank the Reviewer for their time and assessment of our work. We wish to emphasise, as above, that this work is mechanistic, but the mechanisms we interrogate relate to the quantitative flow of biochemical information in the pathway, rather than to any changes in “signalling anatomy”, as in many reductionist studies. Studying quantitative PI3K signalling mechanisms has hitherto not been possible due to technical limitations, and we solve this challenge through the systematic development of three orthogonal workflows and analytical pipelines.*

The need for analytical sophistication

The Reviewer implies that the sophisticated analysis used rather guilds the lily. We would like to rebut that strongly. This analysis is absolutely necessary for accurate analysis of single-cell signalling data. The Reviewer appears not to fully appreciate the premises and power of information theory for handling single-cell data and for capturing the effect of signalling heterogeneity on biochemical information transfer. Information transfer indeed cannot be easily deduced from visual inspection of the data alone, as the traces in Figs. 2-3 represent median trajectories and their confidence intervals; however, for every single one of the underlying time points, we have single-cell distributions from >100 cells (TIRF analyses) and >1000 cells (KTR analyses). This “hidden” single-cell resolution is captured in our calculations of channel capacity and mutual information. Considering the example of two cellular systems that respond to a given stimulus with the same average amplitude (what one would capture through a bulk Western blot analysis, for example) illustrates this, as at the single-cell level, the two systems may exhibit entirely different response distributions (i.e. probabilities). Simply looking at the average responses as in conventional signalling experiments, one would erroneously conclude that the two systems respond identically, yet this would clearly not be the case if studied with single-cell resolution. Information theoretical analyses allow us to take this heterogeneity into account in our quantitative evaluations of signalling fidelity. These concepts and related examples are covered in great detail by (Rhee, Cheong, and Levchenko 2012; Levchenko 2023; Feinberg and Levchenko 2023). We have now added a new paragraph on information theory in the Introduction, to aid readers who may be less familiar with these concepts. We apologise to the Reviewer for not doing this from the start as we think this may have prevented misunderstanding.

Quantitative accuracy based on signalling trajectories

Crucially, recent algorithm development in the field of information theory now allows us to leverage the information contained in the entire signalling time course, not just individual snapshot responses. As shown in Figs. 2B and 2C, this is essential for accurate measurements of signalling fidelity and for estimating the amount of growth factor-specific information that is transduced specifically through the PI3K/AKT pathway. This quantitative metric and our tools can now be used for screening purposes to identify therapeutic modalities that normalise growth factor-specific signaling transfer (note that this is different to the complete ablation of signal transfer that results from usage of high-dose PI3K inhibitors which consequently give rise to substantial toxicity). **In short, our work now makes it possible to realise the potential of PI3K signalling dynamics as a pharmacological target, a concept first proposed by (Behar et al. 2013).** We have now added this key message to the summary paragraph of the Introduction.

Quantitative mechanisms

The Reviewer’s assessment of our work as purely phenomenological rather dismisses what we believe to be key insights made in systems after defined perturbations. Specifically, we provide the first evidence of corrupted signal encoding as the mechanism of action of one of the most common human oncogenes. This is a significant step change in understanding, compared to a wealth of studies which are

predominantly phenomenological and/or treat the pathway output in binary or simple quantitative terms. Our description offers greater explanatory power and the prospect of identifying, and quantifying context-dependent signal transfer with the precision of the physical sciences.

Finally, we fully agree with the Reviewer that the blurring is dependent on the allelic dose of PIK3CA^{H1047R}. This is a key message of the work, supported by data from HeLa and iPSC model systems. It demonstrates the importance of quantitative signalling and cellular engineering approaches for accurate understanding of the mechanism of action of oncogenic PIK3CA mutations.

1. The ph domain analysis described at the beginning of the paper is a bit distracting, and could be described in the supplement only – leaving space for better description of other analysis details (see below).

AUTHORS' RESPONSE:

We thank the Reviewer for this suggestion but would prefer to retain the information on the PH domain workflow in the main text. This is one of the three critical workflows that we systematically optimise to enable the field to dissect quantitative PI3K signalling with high single-cell and temporal precision. For years, the field has been limited by lack of robust benchmarking of existing PH domain-based PI3K reports as well as lack of systematic analytical pipelines to process the resulting data. We worry that burying this information in the supplementary material will limit the adoption of best practice workflows. This technological development, if visible, is also likely to be a major source of subsequent citations as readers become aware of the workflows.

2. The analysis of mathematical information-theoretic analysis of trajectory responses is not well described. The authors just cite a PLoS Comp Biol paper (ref 21). And their conclusion on p4 is puzzling. It does seem that the concentration of IGF1 matters more for degree of PI3K product formation in wild type than seen for EGF. With mutant PIK3CA, though, everything seems the same for IGF1 but EGF still has effects that seem more different for different concentrations and more different for 1x vs. 2x. Surely this should be captured in the consideration of information transfer. And what does it mean? In 2x, the cells seem especially sensitive to the highest EGF dose. A problem is that the analysis is not really explained, so it is difficult to extract the major point intended in figure 2. The authors do not make a convincing case that the mutant PIK3CA 'erodes signaling fidelity'. The cells do seem to fail to distinguish between distinct doses of IGF1, but not so much for EGF based on comparing before/after ligand addition. In fact, the data suggests that the PIK3CA mutation just switches on PI3K, and that EGF can augment this further.

AUTHORS' RESPONSE: *We apologise to the Reviewer for any missing information that appears to have led to misunderstanding of our data analysis and conclusions. We have now added a paragraph on information theory to the Introduction, emphasising key concepts that we also cover in our overall response to Reviewer 3*

(see above). To reiterate, the key misunderstanding of the Reviewer relates to their focusing on the average trajectories without consideration of the underlying single-cell distributions. It is critical to evaluate our conclusions in the context of the exact quantifications in Figs. 2B and 3B and not be misled by the summarised trajectory responses alone. This in itself demonstrates the necessity of the more sophisticated information theoretic analyses of single-cell heterogeneity to avoid erroneous conclusions. We note, however, that application of information theory for accurate understanding of biochemical signalling is not a novel concept; it was pioneered initially by the groups of Andre Levchenko, Alexander Hoffmann and Shinya Kuroda across three independent Science papers in the period 2011-2014 (Cheong et al. 2011; Uda et al. 2013; Selimkhanov et al. 2014). Our contribution now is to make it applicable to PI3K signalling by developing the necessary experimental workflows, further demonstrating how such analyses can change our understanding of one of the most widely studied oncogenic mutations to date.

3. The authors comment on p5 that “the EGF-induced PIP3/PI(3,4)P2 reporter response in mutant cells appeared amplified and largely indistinguishable from that of IGF1 in wild-type cells”. This does not seem to be true. Background is higher, the overshoot is absent, and kinetics look different. Since this observation underlies the hypothesis that PIK3CAH1047R expression may corrupt the cellular ability to resolve different growth factor inputs from one another, the similarity should be assessed in more detail.

AUTHORS’ RESPONSE: *The confusion here relates to the above-mentioned misunderstanding of the information theoretic analyses and the need for taking single-cell distributions into account when evaluating the fidelity of cellular signalling. Our conclusions are supported by the analyses in Figure 2B, which quantify the reduced ability of PIK3CA^{H1047R} cells to sense IGF1 but not EGF, bringing the amount of information that is transferred through the PI3K signalling channel to a similar level for the two growth factors. This initial evidence of “blurring” is then corroborated in independent experiments using an AKT kinase translocation reporter, including direct quantification of the “blurring” based on mutual information (Fig. 3B).*

4. Do the authors know if the overexpressed ph domain used in Figures 1 and 2 sequesters PI3K products and competes with endogenous ph domains?

AUTHORS’ RESPONSE: *We are aware of this concern and addressed it as follows:*

- 1) *We use a minimal amount of PH domain reporter constructs; to achieve low-level yet uniform expression of the reporter, we include an excess of a small carrier plasmid (empty pUC19). We have now made this point explicit in the Methods section and apologise for not having done so from the beginning.*
- 2) *We only use PH domain reporters in transient expression experiments and validate our observations using two orthogonal signalling approaches – KTR-based analysis of AKT activity and mass cytometry-based PTM analysis in unmodified 3D spheroids.*

5. The idea in Figure 3 is interesting, but the authors do not really explain how the calculation of mutual information between IGF1 and each other GF leads

to the conclusion drawn. In wildtype cells, insulin/IGF1 ligands activate PI3K more than EGF ligands do. With mutant PIK3CA, background AKT is elevated and PI3K signaling seems perhaps to be sensitized to EGF/EPI for 1x (not for 2x though). It seems that a more comprehensive analysis of signaling networks is needed to interpret this as a 'blurring'. It would also be valuable to perform parallel analysis for other ligands of other receptors.

AUTHORS' RESPONSE: *We refer the Reviewer to our responses to points 2 and 3, which we hope have now clarified any misunderstandings in relation to application of information theoretic analyses. As mentioned, we now include additional background on this in the Introduction. Of note, network biology cannot be used for evaluation of signal blurring with the quantitative precision afforded by information theory; the latter also does not require inclusion of extensive prior knowledge assumptions, making for a less biased outcome.*

6. Do the authors have controls to ensure that the fixing in the experiment shown in Figure 4 does not affect outcome?

AUTHORS' RESPONSE: *We are not entirely clear which additional controls the Reviewer may refer to as the mass cytometry results are entirely concordant with our observations in live-cell TIRF- and KTR-based analyses of PI3K/AKT signalling. We also present BYL719 as control treatment in our mass cytometry analyses, demonstrating the expected shifts following PI3K inhibition.*

7. The authors state on p6 that PIK3CA mutants yield stronger AKT and ERK phosphorylation responses to EGF. This seems clear for AKT from Figs 4B and S5A, but not really for ERK, unless I am missing something.

AUTHORS' RESPONSE: *We apologise for this oversight and misleading wording. Our reference to EGF-induced ERK phosphorylation was meant to be with respect to the more sustained response in PIK3CA^{H1047R} mutant cells relative to wildtype controls. We have now revised the text from:*

“This resulted in stronger and more sustained responses both at the level of pAKT^{S473} and pERK1/2^{T202/Y204; T185/Y187}.”

to:

“This resulted in stronger and more sustained responses at the level of pAKT^{S473}. Similarly, albeit to a lesser degree, the pERK1/2^{T202/Y204; T185/Y187} response in PIK3CA^{H1047R} cells remained higher than wildtype controls after 30 minutes of EGF stimulation, suggestive of a slower switch-off (Fig. 4B, 4C, S5A).”

8. The heterogeneity argument at the top of p7 seems to apply (with PIK3CA mutants) to AKT, but not ERK.

AUTHORS' RESPONSE: *We agree with the Reviewer but are not entirely sure where this confusion arose as we do not claim increased signalling heterogeneity at the level*

of ERK phosphorylation. To improve clarity, we have now added the bolded text to our paragraph on signalling heterogeneity in the relevant Results section:

“Relative to wild-type controls, the **PI3K pathway** signaling responses in both single- and double-copy *PIK3CA^{H1047R}* mutant HeLa cells also exhibited increased single-cell variability as a function of time and growth factor stimulation, most notably for pAKT^{S473} (**Fig. 4B**). This was reproduced with independent clones (**Fig. S5B**), and in additional dose-response, time course experiments using 1 nM, 10 nM, and 100 nM IGF1 or EGF (**Fig. S6A,B**). We therefore conclude that oncogenic *PIK3CA^{H1047R}* does not simply shift the PI3K/AKT signaling response to a higher mean but also acts to enhance signaling heterogeneity.”

9. The mechanistic details about interactions etc in the second paragraph of Discussion do not seem to be related to the data in the paper.

AUTHORS' RESPONSE: *We respectfully disagree with the Reviewer. We discuss prior knowledge of molecular mechanisms in the context of new observations and hypotheses made possible through our quantitative analyses.*

REFERENCES

- Behar, Marcelo, Derren Barken, Shannon L. Werner, and Alexander Hoffmann. 2013. "The Dynamics of Signaling as a Pharmacological Target." *Cell* 155 (2): 448–61. <https://doi.org/10.1016/j.cell.2013.09.018>.
- Cheong, Raymond, Alex Rhee, Chiaochun Joanne Wang, Ilya Nemenman, and Andre Levchenko. 2011. "Information Transduction Capacity of Noisy Biochemical Signaling Networks." *Science* 334 (6054): 354–58. <https://doi.org/10.1126/science.1204553>.
- Dias, Matheus Henrique, Anoek Friskes, Siying Wang, Joao M. Fernandes Neto, Frank van Gemert, Soufiane Mourragui, Chrysa Papagianni, et al. 2024. "Paradoxical Activation of Oncogenic Signaling as a Cancer Treatment Strategy." *Cancer Discovery* 14 (7): 1276–1301. <https://doi.org/10.1158/2159-8290.CD-23-0216>.
- Feinberg, Andrew P, and Andre Levchenko. 2023. "Epigenetics as a Mediator of Plasticity in Cancer." *Science* 379 (6632).
- Jenkins, Dagan. 2024. "How Do Stochastic Processes and Genetic Threshold Effects Explain Incomplete Penetrance and Inform Causal Disease Mechanisms?" *Philosophical Transactions of the Royal Society B: Biological Sciences* 379 (1900): 20230045. <https://doi.org/10.1098/rstb.2023.0045>.
- Levchenko, Andre. 2023. "Genetic Diseases: How the Noise Fits In." *Current Biology* 33 (6): R228–30. <https://doi.org/10.1016/j.cub.2023.02.052>.
- Opzoomer, James W., Rhianna O'Sullivan, Jahangir Sufi, Ralitsa Madsen, Xiao Qin, Ewa Basiarz, and Christopher J. Tape. 2024. "SIGNAL-Seq: Multimodal Single-Cell Inter- and Intra-Cellular Signalling Analysis." bioRxiv. <https://doi.org/10.1101/2024.02.23.581433>.
- Rhee, Alex, Raymond Cheong, and Andre Levchenko. 2012. "The Application of Information Theory to Biochemical Signaling Systems." *Physical Biology* 9 (4): 045011. <https://doi.org/10.1088/1478-3975/9/4/045011>.
- Selimkhanov, Jangir, Brooks Taylor, Jason Yao, Anna Pilko, John Albeck, Alexander Hoffmann, Lev Tsimring, and Roy Wollman. 2014. "Accurate Information Transmission through Dynamic Biochemical Signaling Networks." *Science* 346 (6215): 1370–73. <https://doi.org/10.1126/science.1254933>.
- Uda, S., T. H. Saito, T. Kudo, T. Kokaji, T. Tsuchiya, H. Kubota, Y. Komori, Y.-i. Ozaki, and S. Kuroda. 2013. "Robustness and Compensation of Information Transmission of Signaling Pathways." *Science* 341 (6145): 558–61. <https://doi.org/10.1126/science.1234511>.

The revised paper by Madsen et al. is largely unchanged from the original submission to Science, and the authors provide a great deal of justification for different elements of the initial paper on a variety of points in their 'Response to Reviewers'. Their core point is that collecting trajectories of responses for multiple individual cells for different ligands and a given ligand at different concentrations reveals that the fidelity of dose-dependent signal transfer (encoding of nature or concentration of ligand) is lost with oncogenic PI3K activation in the contexts used here. This is a useful result, and I think warrants publication in Science Signaling.

One concern I have is that the sophisticated analysis reduces the 'fidelity' to a single number in Figs. 2B and 3B for example, suggesting altered fidelity but giving little sense of what has changed to cause this to be the case. Intuiting what is behind the Information Capacity or Mutual Information will be difficult for readers. It would therefore be helpful if the authors could expand in the Discussion or elsewhere in the manuscript on this sentence in the 'Response to Reviewers': "our tools can now be used for screening purposes to identify therapeutic modalities that normalise growth factor-specific signaling transfer". This could be a key message of the paper, but how to get to this (or to 'target dynamics') is left very vague.

The authors now comment on p3 that "Our work now opens for the possibility of using PI3K signaling dynamics as a pharmacological target, a concept first proposed on theoretical grounds by Behar et al. (19)." Since that Behar paper 11 years ago, this point has been established in several situations. The current work is consistent with the idea, but really does not hint how to go about this. The suggestion is made on p9 that the work: "makes possible the prospective development and application of pharmacological approaches to tune pathological PI3K signaling responses back to normal, for example through allosteric modulation of receptor-specific coupling mechanisms." Unfortunately no ideas are given on how to do this, which does limit the point's impact. The Discussion as currently written does not illustrate the authors' thinking in this sense.

AUTHORS' RESPONSE: We thank the Reviewer for suggesting that we expand on one of our key points in the Discussion. We have now included the red text below to add more specific detail on the types of mechanisms that warrant further investigation in the context of effective tuning/normalisation of growth factor-specific signalling responses in cells with oncogenic PIK3CA.

Finally, our quantitative PI3K signaling framework can now be used for screening purposes to identify therapeutic modalities that normalize growth factor-specific signal transfer in PIK3CA^{H1047R} cells. This would be different from direct PI3K α inhibition which, as suggested by our genetic and pharmacological data, does not restore normal signaling dynamics. Given

our evidence for conserved receptor-specific mechanisms of PI3K signal encoding, alternative pharmacological approaches may instead feature modulation of receptor-specific adaptor proteins, phosphatases and/or E3 ligases, all of which are critical for dynamic signaling control through regulation of receptor internalization, recycling and degradation (8). Similar modulation has also been suggested for RAS/MAPK signaling (58), in light of the finding that oncogenic mutations in this pathway also remain dependent on upstream growth factor inputs yet fail to transmit these reliably (9). Given the likely dependence on a direct PI3K α -RAS interactions for the EGF response amplification in *PIK3CA*^{H1047R} cells, the PI3K α -RAS breaker (59) may be an excellent candidate for testing of quantitative, growth factor-specific PI3K signaling dynamics as a pharmacological target.

31st Oct 2024

Manuscript Number: MSB-2024-12712-T

Title: Oncogenic PIK3CA corrupts growth factor signaling specificity

Author: Bart Vanhaesebroeck

Ralitsa Radostinova Madsen

Dear Dr. Radostinova Madsen,

Thank you for the submission of your revised manuscript to Molecular Systems Biology. I am pleased to inform you that we will be able to accept your manuscript pending the following final amendments:

- 1) Please download the EMBO Press "Author Checklist" and complete all relevant questions. This file should be uploaded with your submission. This file can be downloaded from our website at:
<https://www.embopress.org/page/journal/17444292/authorguide>
- 2) Please ensure that all authors in the manuscript file are included in the submission system, as currently some authors are missing.
- 3) Please upload the manuscript as a .docx file, with the figures removed (see instructions below for uploading figures), and no track changes.
- 4) In the main manuscript file, please include keywords to max. 5.
- 5) Please rename your 'Data and materials availability' statement to 'Data availability'. Please also ensure that you include the information for the raw sequencing data available in GEO. The data availability section should be formatted according to the example below:
"The datasets and computer code produced in this study are available in the following databases:
- Chip-Seq data: Gene Expression Omnibus GSE46748 (<https://www.ncbi.nlm.nih.gov/geo/query/acc.cgi?acc=GSE46748>)
- Modeling computer scripts: GitHub (<https://github.com/SysBioChalmers/GECKO/releases/tag/v1.0>)
- [data type]: [full name of the resource] [accession number/identifier] ([doi or URL or identifiers.org/DATABASE:ACCESSION])"
- 6) Please rename "Competing Interests" to "Disclosure and competing interests statement". We updated our journal's competing interests policy in January 2022 and request authors to consider both actual and perceived competing interests. Please review the policy <https://www.embopress.org/competing-interests> and update your competing interests if necessary.
- 7) Author contributions: Please remove it from the manuscript and specify author contributions in our submission system. CRediT has replaced the traditional author contributions section because it offers a systematic machine-readable author contributions format that allows for more effective research assessment. You are encouraged to use the free text boxes beneath each contributing author's name to add specific details on the author's contribution. More information is available in our guide to authors:
<https://www.embopress.org/page/journal/17574684/authorguide#authorshipguidelines>
- 8) References: Please correct the reference citation in the reference list. References should be alphabetical, and where there are more than 10 authors on a paper, please only list the first 10 followed by "et al.". Please also rename this section to 'References' instead of 'References and Notes'. Please check "Author Guidelines" for more information.
<https://www.embopress.org/page/journal/17574684/authorguide#referencesformat>
- 9) Our journal encourages inclusion of *data citations in the reference list* to directly cite datasets that were re-used and obtained from public databases. Data citations in the article text are distinct from normal bibliographical citations and should directly link to the database records from which the data can be accessed. In the main text, data citations are formatted as follows: "Data ref: Smith et al, 2001" or "Data ref: NCBI Sequence Read Archive PRJNA342805, 2017". In the Reference list, data citations must be labeled with "[DATASET]". A data reference must provide the database name, accession number/identifiers and a resolvable link to the landing page from which the data can be accessed at the end of the reference. Further instructions are available at .
- 10) In the Methods, please take care of the following:
 - The Materials and Methods section should be renamed to "Methods".
 - Studies with human research participants: The use of human iPSCs may require ethics approval (e.g. IRB) and informed consent. If the need for approval is not required, please cite the reason (e.g. non-human subject research because the iPSCs used were de-identified/coded with no identifying information).
 - Studies with cells derived from animals: The establishment of MEFs from mice may require ethics approval, if the MEFs were established for the purpose of this study. Please clarify whether or not animals were used in the course of the study.
- 11) All Materials and Methods need to be described in the main text using our 'Structured Methods' format. According to this format, the Methods section includes a Reagents and Tools Table (listing key reagents, experimental models, software and relevant equipment and including their sources and relevant identifiers) followed by a Methods and Protocols section describing the methods, ideally using a step-by-step protocol format. The aim is to facilitate adoption of the methodologies across labs. Please download and fill our Reagents and Tools Table template (.docx), which you can find in our author guidelines:
<https://www.embopress.org/page/journal/14693178/authorguide#structuredmethods>.

An example of a Method paper with Structured Methods can be found here:
<https://www.embopress.org/doi/10.15252/msb.20178071>. "

12) Please place individual sections of the manuscript in the following order: Title page - Abstract & Keywords - Introduction - Results - Discussion - Methods - Data Availability - Acknowledgements - Disclosure and Competing Interests Statement - References - Figure Legends - Expanded View Figure Legends. "One-Sentence Summary" and "Supplementary Materials" sections should be removed.

13) For the figures and figure legends, please take care of the following:

- Please remove all figures from main manuscript file and leave only main figure legends placed after the references. Main figures should be uploaded as individual, high-resolution files. Regarding the supplementary figures, you can upload up to 5 as Expanded View (EV) Figures (EV figures will be displayed in the main HTML of the paper in a collapsible format). EV figures need to be uploaded as separate Figure files as well, with their legends in the manuscript file, after the main figure legends, and renamed with the following nomenclature 'Figure EV1'. The remaining figures should be compiled in one PDF file labeled "Appendix" with their legends. Please ensure that the figure legends are included with the figures in the appendix, and that the appendix has a table of contents with page numbers. Please also ensure that the nomenclature for the figures and tables is correct, i.e. "Appendix Figure S1" and "Appendix Table S1". Please check "Author Guidelines" for more information:
<https://www.embopress.org/page/journal/17574684/authorguide#figureformat>

- Please note that the exact p values are not provided in the legends of figures 5c-d.

- Please note that the box plots need to be defined in terms of minima, maxima, centre, bounds of box and whiskers, and percentile in the legends of figures 5c-d.

- Please note that information related to n is missing in the legend of figure 2b.

- Please note that the measure of center for the error bars needs to be defined in the legend of figure 2b.

14) Tables: Tables S1-S4 can be either included in the main manuscript file as Tables, placed between main and EV figure legends with the nomenclature Table 1-4, or they can remain supplementary tables and be compiled in the Appendix PDF with the nomenclature Appendix Table S1-S4. Please ensure that the appropriate callouts for these tables is given throughout the manuscript.

15) Funding: Please ensure that all funding sources are entered into the manuscript submission system.

16) Synopsis:

- Synopsis image: Please provide a graphic that summarises the main findings of the manuscript on a glance and upload it as a high-resolution jpeg file 550 pixels wide x (300-600) pixels high.

- Synopsis text: Please provide a short standfirst (maximum of 300 characters, including space), limit the bullet points to max. 5 and upload it as a separate .doc file. Please write the bullet points to summarise the key NEW findings. They should be designed to be complementary to the abstract - i.e. not repeat the same text. We encourage inclusion of key acronyms and quantitative information (maximum of 30 words / bullet point). Please use the passive voice.

17) Source Data: Our colleague Hannah Sonntag will contact you separately regarding any requests for Source Data along with a checklist. Please ensure that a completed Source Data checklist is uploaded with your resubmission, along with a single source data file (zipped) per figure, with the panels clearly visible in the folder structure.

18) As part of the EMBO Publications transparent editorial process initiative (see our policy here:

https://www.embopress.org/transparent-process#Review_Process), Molecular Systems Biology will publish online a Peer Review File (PRF) to accompany accepted manuscripts. This file will be published in conjunction with your paper and will include the anonymous referee reports, your point-by-point response and all pertinent correspondence relating to the manuscript. Let us know whether you agree with the publication of the PRF and as here, if you want to remove or not any figures from it prior to publication. Please note that the Authors checklist will be published at the end of the PRF.

19) Please provide a point-by-point letter INCLUDING my comments as well as your detailed responses (as Word file).

I look forward to reading a new revised version of your manuscript as soon as possible.

Yours sincerely,

Poonam Bheda, PhD
Scientific Editor
Molecular Systems Biology

School of Life Sciences
University of Dundee

MRC Protein Phosphorylation and Ubiquitylation Unit

Dr. Ralitsa R. Madsen
t: +44 (0) 7871688829
e: rmadsen001@dundee.ac.uk

06 November 2024

Dr. Poonam Bheda, PhD
Scientific Editor
Molecular Systems Biology

Dear Dr. Bheda,

We are pleased to submit the requested revision of Molecular Systems Biology manuscript MSB-2024-12712. I hope we have addressed editorial comments and requests to your satisfaction. The following pages contain a point-by-point response to all editorial comments (bolded for clarity).

Please do not hesitate to get in touch if anything is missing.

We thank you for your professional handling of our manuscript.

Kindest regards

Ralitsa Madsen on behalf of all authors

**1) Please download the EMBO Press "Author Checklist" and complete all relevant questions. This file should be uploaded with your submission. This file can be downloaded from our website at:
<https://www.embopress.org/page/journal/17444292/authorguide>**

This has now been done. According to the Checklist, we have also added additional details to our Methods regarding any exclusion criteria in TIRF and KTR image analyses.

2) Please ensure that all authors in the manuscript file are included in the submission system, as currently some authors are missing.

This has now been done.

3) Please upload the manuscript as a .docx file, with the figures removed (see instructions below for uploading figures), and no track changes.

This has now been done.

4) In the main manuscript file, please include keywords to max. 5.

This has now been done.

5) Please rename your 'Data and materials availability' statement to 'Data availability'. Please also ensure that you include the information for the raw sequencing data available in GEO. The data availability section should to be formatted according to the example below:

"The datasets and computer code produced in this study are available in the following databases:

**- Chip-Seq data: Gene Expression Omnibus GSE46748
(<https://www.ncbi.nlm.nih.gov/geo/query/acc.cgi?acc=GSE46748>)**

**- Modeling computer scripts: GitHub
(<https://github.com/SysBioChalmers/GECKO/releases/tag/v1.0>)**

- [data type]: [full name of the resource] [accession number/identifier] ([doi or URL or identifiers.org/DATABASE:ACCESSION])"

This has now been done.

**6) Please rename "Competing Interests" to "Disclosure and competing interests statement". We updated our journal's competing interests policy in January 2022 and request authors to consider both actual and perceived competing interests. Please review the policy
<https://www.embopress.org/competing-interests> and update your competing interests if necessary.**

This has now been done.

7) Author contributions: Please remove it from the manuscript and specify

author contributions in our submission system. CRediT has replaced the traditional author contributions section because it offers a systematic machine-readable author contributions format that allows for more effective research assessment. You are encouraged to use the free text boxes beneath each contributing author's name to add specific details on the author's contribution. More information is available in our guide to authors: <https://www.embopress.org/page/journal/17574684/authorguide#authorshipguidelines>

This has now been done.

8) References: Please correct the reference citation in the reference list. References should be alphabetical, and where there are more than 10 authors on a paper, please only list the first 10 followed by "et al.". Please also rename this section to 'References' instead of 'References and Notes'. Please check "Author Guidelines" for more information.

<https://www.embopress.org/page/journal/17574684/authorguide#referencesformat>

This has now been done.

9) Our journal encourages inclusion of *data citations in the reference list* to directly cite datasets that were re-used and obtained from public databases. Data citations in the article text are distinct from normal bibliographical citations and should directly link to the database records from which the data can be accessed. In the main text, data citations are formatted as follows: "Data ref: Smith et al, 2001" or "Data ref: NCBI Sequence Read Archive PRJNA342805, 2017". In the Reference list, data citations must be labeled with "[DATASET]". A data reference must provide the database name, accession number/identifiers and a resolvable link to the landing page from which the data can be accessed at the end of the reference. Further instructions are available at

<https://www.embopress.org/page/journal/17574684/authorguide#referencesformat>.

We do not reuse previously generated datasets.

10) In the Methods, please take care of the following:

- The Materials and Methods section should be renamed to "Methods".

This has now been done.

- Studies with human research participants: The use of human iPSCs may require ethics approval (e.g. IRB) and informed consent. If the need for approval is not required, please cite the reason (e.g. non-human subject research because the iPSCs used were de-identified/coded with no identifying information).

The iPSCs were generated and described in detail in a previous study (Madsen et al. 2019), using a commercially available parental iPSC line. This has now been made explicit in the Methods and the accompanying Reagents and Tools Table.

- Studies with cells derived from animals: The establishment of MEFs from mice may require ethics approval, if the MEFs were established for the purpose of this study. Please clarify whether or not animals were used in the course of the study.

The immortalized MEFs used in this study were generated and banked as part of a previous study (Foukas et al. 2010) from the Vanhaesebroeck Group. This has now been made clearer to readers in the Methods and the accompanying Reagents and Tools Table.

11) All Materials and Methods need to be described in the main text using our 'Structured Methods' format. According to this format, the Methods section includes a Reagents and Tools Table (listing key reagents, experimental models, software and relevant equipment and including their sources and relevant identifiers) followed by a Methods and Protocols section describing the methods, ideally using a step-by-step protocol format. The aim is to facilitate adoption of the methodologies across labs. Please download and fill our Reagents and Tools Table template (.docx), which you can find in our author guidelines: <https://www.embopress.org/page/journal/14693178/authorguide#structuredmethods>.

An example of a Method paper with Structured Methods can be found here: <https://www.embopress.org/doi/10.15252/msb.20178071>. "

This has now been done. For step-by-step protocols, we provide direct links to the publicly deposited versions on protocols.io.

12) Please place individual sections of the manuscript in the following order: Title page - Abstract & Keywords - Introduction - Results - Discussion - Methods - Data Availability - Acknowledgements - Disclosure and Competing Interests Statement - References - Figure Legends - Expanded View Figure Legends. "One-Sentence Summary" and "Supplementary Materials" sections should be removed.

This has now been done.

13) For the figures and figure legends, please take care of the following:
- Please remove all figures from main manuscript file and leave only main figure legends placed after the references. Main figures should be uploaded as individual, high-resolution files. Regarding the supplementary figures, you can upload up to 5 as Expanded View (EV) Figures (EV figures will be displayed in the main HTML of the paper in a collapsible format). EV figures need to be uploaded as separate Figure files as well, with their legends in

the manuscript file, after the main figure legends, and renamed with the following nomenclature 'Figure EV1'. The remaining figures should be compiled in one PDF file labeled "Appendix" with their legends. Please ensure that the figure legends are included with the figures in the appendix, and that the appendix has a table of contents with page numbers. Please also ensure that the nomenclature for the figures and tables is correct, i.e. "Appendix Figure S1" and "Appendix Table S1". Please check "Author Guidelines" for more information:

<https://www.embopress.org/page/journal/17574684/authorguide#figureformat>

This has now been done. Note also that we have added a statement regarding usage of Biorender for Figure 6C, including a publication license link under CC-BY terms (added to the figure legend).

- Please note that the exact p values are not provided in the legends of figures 5c-d.

This has now been added.

- Please note that the box plots need to be defined in terms of minima, maxima, centre, bounds of box and whiskers, and percentile in the legends of figures 5c-d.

This has now been done.

- Please note that information related to n is missing in the legend of figure 2b.

This has now been added.

- Please note that the measure of center for the error bars needs to be defined in the legend of figure 2b.

This has now been done.

14) Tables: Tables S1-S4 can be either included in the main manuscript file as Tables, placed between main and EV figure legends with the nomenclature Table 1-4, or they can remain supplementary tables and be compiled in the Appendix PDF with the nomenclature Appendix Table S1-S4. Please ensure that the appropriate callouts for these tables is given throughout the manuscript.

These tables are now included in the Appendix and references as specified above.

15) Funding: Please ensure that all funding sources are entered into the manuscript submission system.

This has now been done.

16) Synopsis:

- **Synopsis image:** Please provide a graphic that summarises the main findings of the manuscript on a glance and upload it as a high-resolution jpeg file 550 pixels wide x (300-600) pixels high.
- **Synopsis text:** Please provide a short standfirst (maximum of 300 characters, including space), limit the bullet points to max. 5 and upload it as a separate .doc file. Please write the bullet points to summarise the key **NEW** findings. They should be designed to be complementary to the abstract - i.e. not repeat the same text. We encourage inclusion of key acronyms and quantitative information (maximum of 30 words / bullet point). Please use the passive voice.
- **Please check your synopsis text and image before submission with your revised manuscript. Please be aware that in the proof stage minor corrections only are allowed (e.g., typos).**

This has now been done. Note also that we have shortened the Abstract itself due to word limitations in the submission system.

17) Source Data: Our colleague Hannah Sonntag will contact you separately regarding any requests for Source Data along with a checklist. Please ensure that a completed Source Data checklist is uploaded with your resubmission, along with a single source data file (zipped) per figure, with the panels clearly visible in the folder structure.

This has now been done.

18) As part of the EMBO Publications transparent editorial process initiative (see our policy here: https://www.embopress.org/transparent-process#Review_Process), Molecular Systems Biology will publish online a Peer Review File (PRF) to accompany accepted manuscripts. This file will be published in conjunction with your paper and will include the anonymous referee reports, your point-by-point response and all pertinent correspondence relating to the manuscript. Let us know whether you agree with the publication of the PRF and as here, if you want to remove or not any figures from it prior to publication. Please note that the Authors checklist will be published at the end of the PRF.

We agree to the publication of the PRF as is.

19) Please provide a point-by-point letter INCLUDING my comments as well as your detailed responses (as Word file).

This has now been done.

11th Nov 2024

Manuscript number: MSB-2024-12712R

Title: Oncogenic PIK3CA corrupts growth factor signaling specificity

Dear Dr Radostinova Madsen,

Thank you again for sending us your revised manuscript. We are now satisfied with the modifications made and I am pleased to inform you that your paper has been accepted for publication.

Yours sincerely,

Sincerely,

Poonam Bheda, PhD
Scientific Editor
Molecular Systems Biology
